# SYNAPTIC WEIGHT DISTRIBUTIONS DEPEND ON THE GEOMETRY OF PLASTICITY

**Roman Pogodin**[*]
McGill & Mila
`roman.pogodin@mila.quebec`

**Jonathan Cornford**[*]
McGill & Mila
`cornforj@mila.quebec`

**Arna Ghosh**
McGill & Mila

**Gauthier Gidel**
Université de Montréal[1] & Mila

**Guillaume Lajoie**
Université de Montréal[2] & Mila

**Blake Aaron Richards**
McGill[3], Mila & CIFAR[4]

## ABSTRACT

A growing literature in computational neuroscience leverages gradient descent and learning algorithms that approximate it to study synaptic plasticity in the brain. However, the vast majority of this work ignores a critical underlying assumption: the choice of distance for synaptic changes – i.e. the geometry of synaptic plasticity. Gradient descent assumes that the distance is Euclidean, but many other distances are possible, and there is no reason that biology necessarily uses Euclidean geometry. Here, using the theoretical tools provided by mirror descent, we show that the distribution of synaptic weights will depend on the geometry of synaptic plasticity. We use these results to show that experimentally-observed log-normal weight distributions found in several brain areas are not consistent with standard gradient descent (i.e. a Euclidean geometry), but rather with non-Euclidean distances. Finally, we show that it should be possible to experimentally test for different synaptic geometries by comparing synaptic weight distributions before and after learning. Overall, our work shows that the current paradigm in theoretical work on synaptic plasticity that assumes Euclidean synaptic geometry may be misguided and that it should be possible to experimentally determine the true geometry of synaptic plasticity in the brain.

## 1 INTRODUCTION

Many computational neuroscience studies use gradient descent to train models that are then compared to the brain Schrimpf et al. (2018); Nayebi et al. (2018); Yamins et al. (2014); Bakhtiari et al. (2021); Flesch et al. (2022), and many others explore how synaptic plasticity could approximate gradient descent Richards et al. (2019). One aspect of this framework that is often not explicitly considered is that in order to follow the gradient of a loss in the synaptic weight space, one must have a means of measuring distance in the synaptic space, i.e. of determining what constitutes a large versus a small change in the weights Carlo Surace et al. (2018). In other words, whenever we build a neural network model we are committing to a synaptic geometry. Standard gradient descent assumes Euclidean geometry, meaning that distance in synaptic space is equivalent to the L2-norm of weight changes.

There are however other distances that can be used. For example, in natural gradient descent the Kullback-Leibler divergence of the network's output distributions is used as the distance Martens (2020). More broadly, the theory of mirror descent provides tools for analyzing and building algorithms that use different distances in parameter space Nemirovskij & Yudin (1983); Beck & Teboulle (2003), which has proven useful in a variety of applications Shalev-Shwartz et al. (2012); Bubeck et al. (2015); Lattimore & Gyorgy (2021). However, within computational neuroscience the question of synaptic geometry is often overlooked: most models use Euclidean geometry without considering other options Carlo Surace et al. (2018). This assumption has no basis in neuroscience data, so how could we possibly determine the synaptic geometry used by the brain?

---

[*]Equal contribution. [1]Dept. CS and OR; [2]Dept. Maths and Stats; [3]Dept. of Neurology & Neurosurgery, School of Computer Science, Montreal Neurological Institute; [4]Learning in Machines and Brains Program.

Here, using tools from mirror descent Nemirovskij & Yudin (1983); Beck & Teboulle (2003), we show that synaptic geometry can be determined by observing the distribution of synaptic weight changes during learning. Specifically, we prove that in situations where synaptic changes are relatively small the distribution of synaptic weights depends on the synaptic geometry (with mild assumptions about the loss function and the dataset). We use this result to show that the geometry defines a dual space in which the total synaptic changes are Gaussian. As a result, if one can find a dual space in which experimentally observed synaptic changes are Gaussian, then one knows the synaptic geometry. Applying this framework to existing neural data, which suggests that synaptic weights are log-normally distributed Song et al. (2005); Loewenstein et al. (2011); Melander et al. (2021); Buzsáki & Mizuseki (2014), we conclude that the brain is unlikely to use a Euclidean synaptic geometry. Moreover, we show how to use our findings to make experimental predictions. In particular, we show that it should be possible to use experimentally observed weight distributions before and after learning to rule out different candidate geometries. Altogether, our work provides a novel theoretical insight for reasoning about the learning algorithms of the brain.

## 1.1 RELATED WORK

There is a large and growing literature on approximating gradient-based learning in the brain Lillicrap et al. (2016); Liao et al. (2016); Akrout et al. (2019); Podlaski & Machens (2020); Clark et al. (2021). The vast majority of this work assumes Euclidean synaptic geometry Carlo Surace et al. (2018). Notably, even if the brain does not estimate gradients directly, as long as synaptic weight updates are relatively small, then the brain's learning algorithm must be non-orthogonal to some gradient in expectation Richards & Kording (2023). As such, our work is relevant to neuroscience regardless of the specific learning algorithm used by the brain.

Our work draws strongly from the rich and long-standing literature on mirror descent, which was originally introduced 40 years ago for convex optimization Nemirovskij & Yudin (1983). Recent years have seen a lot of work in this area Beck & Teboulle (2003); Duchi et al. (2010); Bubeck et al. (2015); Ghai et al. (2020); Lattimore & Gyorgy (2021), especially in online optimization settings such as bandit algorithms Shalev-Shwartz et al. (2012); Lattimore & Szepesvári (2020). More recently, researchers have started to apply mirror descent to deep networks and have used it to try to develop better performing algorithms than gradient descent Azizan & Hassibi (2018); Azizan et al. (2021). This work is related to natural gradient descent which also explores non-Euclidean geometries Amari (1985; 1998); Ollivier et al. (2017).

Finally, our work is relevant to experimental neuroscience literature on synaptic weight distributions. Using a variety of techniques, including patch clamping Song et al. (2005) and fluorescent imaging Melander et al. (2021); Loewenstein et al. (2011); Vardalaki et al. (2022), neuroscientists have explored the distributions of synaptic strengths across a variety of brain regions and species. This work has generally reported log-normal distributions in synaptic weights Song et al. (2005); Loewenstein et al. (2011); Melander et al. (2021), though, a recent study observed a more complicated bimodal distribution in log-space Dorkenwald et al. (2022). Moreover, the parameters of log-normal distributions observed in primary auditory cortex Levy & Reyes (2012) are close to optimal in terms of perceptron capacity Zhong et al. (2022). Our work connects this experimental literature to our theoretical understanding of learning in the brain – it makes it possible to test theories of synaptic geometry using weight distribution data.

## 2 MIRROR DESCENT FRAMEWORK

In order to derive our core results we will rely on tools from mirror descent. To introduce it, we first revisit gradient descent. Assume we have a loss function $l(\mathbf{w})$. In gradient descent, we choose the next point in weight space $\mathbf{w}^{t+1}$ from the current point $\mathbf{w}^t$ by minimizing a linearized version of $l(\mathbf{w})$ with a penalty for taking large steps in weight space (where the strength of the penalty is controlled by the learning rate $\eta$). Importantly, penalizing large steps in weight space necessitates a distance function. If we choose the squared $L^2$-norm as our distance function we obtain:

$$\mathbf{w}^{t+1} = \arg\min_{\mathbf{w}} g(\mathbf{w}, \mathbf{w}^t), \quad g(\mathbf{w}, \mathbf{w}^t) = l(\mathbf{w}^t) + \nabla l(\mathbf{w}^t)^\top (\mathbf{w} - \mathbf{w}^t) + \frac{1}{2\eta} \|\mathbf{w} - \mathbf{w}^t\|_2^2. \quad (1)$$

This choice of distance function results in the standard gradient descent update when we solve the unconstrained optimization problem in Eq. (1):

$$\mathbf{w}^{t+1} = \mathbf{w}^t - \eta \, \nabla l(\mathbf{w}^t). \tag{2}$$

In mirror descent, we consider a more general set of distance functions provided by the Bregman divergence $D_\phi(\mathbf{w}, \widetilde{\mathbf{w}})$ Bregman (1967). For a strictly convex function, called a *potential*, $\phi(\mathbf{w})$:

$$D_\phi(\mathbf{w}, \widetilde{\mathbf{w}}) = \phi(\mathbf{w}) - \phi(\widetilde{\mathbf{w}}) - \nabla \phi(\widetilde{\mathbf{w}})^\top (\mathbf{w} - \widetilde{\mathbf{w}}). \tag{3}$$

Re-writing Eq. (1) into this more general form we get:

$$\mathbf{w}^{t+1} = \arg\min_{\mathbf{w}} g_\phi(\mathbf{w}, \mathbf{w}^t), \quad g_\phi(\mathbf{w}, \mathbf{w}^t) = l(\mathbf{w}^t) + \nabla l(\mathbf{w}^t)^\top (\mathbf{w} - \mathbf{w}^t) + \frac{1}{\eta} D_\phi(\mathbf{w}, \mathbf{w}^t). \tag{4}$$

We can express the update in closed form using the gradient of the potential:

$$\nabla \phi(\mathbf{w}^{t+1}) = \nabla \phi(\mathbf{w}^t) - \eta \, \nabla l(\mathbf{w}^t). \tag{5}$$

As $\phi$ is strictly convex, $\nabla \phi$ is invertible, so:

$$\mathbf{w}^{t+1} = \nabla \phi^{-1} \left( \nabla \phi(\mathbf{w}^t) - \eta \, \nabla l(\mathbf{w}^t) \right). \tag{6}$$

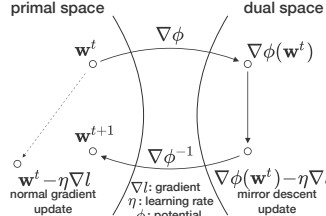

The update moves $\mathbf{w}$ from the "primal" space to the "dual" via $\nabla \phi(\mathbf{w})$, performs a regular gradient descent update in the dual space, and then projects back to the primal space (Fig. 1). We consider $p$-norms $\phi(\mathbf{w}) = \frac{1}{p}\|\mathbf{w}\|_p^p$ and negative entropy

Figure 1: Mirror descent dynamics.

$\phi(\mathbf{w}) = \sum_i |w_i| \log |w_i|$. The former for $p = 2$ recovers standard gradient descent. The latter leads to the exponentiated gradient algorithm Kivinen & Warmuth (1997) ($\odot$ denotes element-wise product):

$$\mathbf{w}^{t+1} = \mathbf{w}^t \odot \, e^{\left( -\eta \nabla l(\mathbf{w}^t) \odot \text{sign} \, \mathbf{w}^t \right)}, \tag{7}$$

Fig. 2 shows how different potentials change problem geometry and, therefore, solutions.

Equation (5) already predicts our main result: if updates in the dual space are approximately independent, or contain noise, over time the sum of the updates will look Gaussian by the central limit theorem (since they sum linearly in the dual space). In fact, Loewenstein et al. (2011) showed that synaptic weights (of rodent auditory cortex) stay log-normal over time (implying Gaussian changes in the log space) and follow multiplicative dynamics in synaptic weight space. This is consistent with the negative entropy potential and in particular Eq. (7). This data, however, wasn't collected when training mice on a specific task. It is not clear if learning-driven dynamics follows the same update geometry. Here, we develop a theory for distinguishing synaptic geometries during training.

Eq. (6) allows us to understand synaptic weight updates via two independent terms: a credit signal and an intrinsic synaptic geometry. The credit signal is the gradient $\nabla l(\mathbf{w}^t)$ – or a more biologically plausible approximation. The synaptic geometry is determined by the potential $\phi$; it captures how the credit signal $\nabla l(\mathbf{w}^t)$ is used to update synaptic weights. Note however, we are not necessarily proposing that this separation is reflected in underlying neurobiological processes, just that it can be used as a model for understanding weight updates. Most studies of biologically plausible deep learning Lillicrap et al. (2016); Liao et al. (2016); Akrout et al. (2019); Podlaski & Machens (2020); Clark et al. (2021) use Euclidean distance by default, effectively ignoring

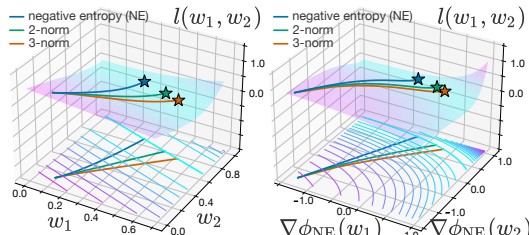

Figure 2: Eq. (4) dynamics for $l(w_1, w_2) = (\frac{1}{2}w_1 + w_2 - 1)e^{\frac{1}{2}w_1 + w_2}$. Blue: negative entropy (NE, Eq. (7)); green: gradient descent/2-norm (Eq. (2)); orange: 3-norm. Left: loss surface (and level sets shown below it) and dynamics in the regular $(w_1, w_2)$ coordinates. Right: same, but in the $(\nabla\phi_{\text{NE}}(w_1), \nabla\phi_{\text{NE}}(w_2))$ coordinates.

the second term (although see Carlo Surace et al. (2018) and Schwarz et al. (2021)). Here we concentrate on the second term, and derive results that are independent of the first term. Therefore, and crucially, previous theoretical studies that derive biologically plausible estimates of credit signals are fully compatible with our work.

## 2.1 Implicit bias in mirror descent

Gradient descent in overparametrized linear regression finds the minimum norm solution: a set of weights $\mathbf{w}$ closest to the initial weights $\mathbf{w}^0$, according to the 2-norm distance Gunasekar et al. (2017); Zhang et al. (2021). This bias towards small changes is referred to as the *implicit bias* of the algorithm. Recently, a similar result was obtained for mirror descent, wherein the implicit bias will depend on the potential $\phi$ Gunasekar et al. (2018). Here we discuss this result and its applicability to gradient-based learning in deep networks.

**Linear regression.** To begin, consider a linear problem: predict $y^n$ from $\mathbf{x}^n$ as $\widehat{y}^n = (\mathbf{x}^n)^\top \mathbf{w}$. For $N$ points and $D$-dimensional weights, we write this as $\mathbf{y} = \mathbf{X}\mathbf{w}$. Gunasekar et al. (2018) showed that for losses $l(\widehat{y}, y)$ with a unique finite root ($l(\widehat{y}, y) \to 0$ iff $\widehat{y} \to y$), the mirror descent solution $\mathbf{w}^\infty$ (assuming it exists) is the closest $\mathbf{w}$ to the initial weights $\mathbf{w}^0$ w.r.t. the Bregman divergence $D_\phi$:

$$\mathbf{w}^\infty = \underset{\mathbf{w}: \, \mathbf{y} = \mathbf{X}^\top \mathbf{w}}{\arg\min} \; D_\phi(\mathbf{w}, \mathbf{w}^0). \tag{8}$$

An example of such a setup is the MSE loss $l(\widehat{y}, y) = (\widehat{y} - y)^2/2$ and an overparametrized model with $D \geq N$. For a fixed $y$, this gives rise to an anti-Hebbian gradient $-(\hat{y} - y)\mathbf{x}$.

**Deep networks.** A crucial condition is hidden in the proof of the above result: the gradient updates $\nabla l(\widehat{y}^n, y^n)$ have to span the space of $\mathbf{x}^n$. For a linear model $\widehat{y} = \mathbf{x}^\top \mathbf{w}$, this is naturally satisfied as $\nabla l(\widehat{y}^n, y^n) = \frac{\partial l(\widehat{y}^n, y^n)}{\partial \widehat{y}^n} \mathbf{x}^n$. For deep networks, Azizan et al. (2021) showed a similar to Eq. (8) result for solutions close to the initial weights, and empirically noted that different potentials result in different final weights. Thus, we can make a similar assumption: if the solution is close to the initial weights, then we can linearize a given deep network $f(\mathbf{w}, \mathbf{x})$ around the initial weights $\mathbf{w}^0$:

$$f^{\text{lin}}(\mathbf{w}, \mathbf{x}, \mathbf{w}^0) = f(\mathbf{w}^0, \mathbf{x}) + \nabla f(\mathbf{w}^0, \mathbf{x})^\top (\mathbf{w} - \mathbf{w}^0). \tag{9}$$

This function is linear in the weights $\mathbf{w}$, but not in the inputs $\mathbf{x}$. Intuitively, if $\mathbf{w}$ doesn't change a lot from $\mathbf{w}^0$, a linearized network should behave similarly to the original one with respect to weight changes. The linear approximation moves us back to a setting akin to linear regression except that the gradients now span $\nabla f(\mathbf{w}^0, \mathbf{x}_i)$ rather than $\mathbf{x}_i$. Thus, for linearized networks, Eq. (8) becomes:

$$\mathbf{w}^\infty = \underset{\mathbf{w}: \, \mathbf{y} = f^{\text{lin}}(\mathbf{w}, \mathbf{x}, \mathbf{w}^0)}{\arg\min} \; D_\phi(\mathbf{w}, \mathbf{w}^0). \tag{10}$$

One may wonder whether it is appropriate to assume that a solution exists near the initial weights when considering real brains. There are two reasons that we argue this is appropriate in our context. First, in biology, animals are never true "blank slates", they instead come with both life experience and evolutionary priors. As such, it is not unreasonable to think that in real brains the solution in synaptic weight space often sits close to the initial weights, and moreover, there would be strong evolutionary pressure for this to be so. Second, neural tangent kernel (NTK; Jacot et al. (2018)) theory shows that, with some assumptions, infinite width networks are identical to their linearized versions, and for finite width networks, the linear approximation gets better as width increases Lee et al. (2019) (although learning dynamics don't always follow NTK theory, see e.g. Bordelon & Pehlevan (2022)). Thus, for very large networks (such as the mammalian brain) it is not unreasonable to think that a linear approximation may be appropriate.

Below, we will develop theory for Eq. (8) (linear regression), and then experimentally show that the results hold for fine-tuning of deep networks (which are close to linearized networks in Eq. (10)).

## 3 Weight distributions in mirror descent

Our goal in this section is to derive a solution for the distribution of the final synaptic weights $\mathbf{w}^\infty$ as a function of the potential $\phi$. To do this, we begin by noting that Eq. (8) can be solved using Lagrange multipliers $\lambda \in \mathbb{R}^N$ that enforce $\mathbf{y} = \mathbf{X}\mathbf{w}$; the Lagrangian is:

$$L(\mathbf{w}, \lambda) = D_\phi(\mathbf{w}, \mathbf{w}_0) + (\mathbf{y} - \mathbf{X}\mathbf{w})^\top \lambda. \tag{11}$$

Solve for the minimum of $L$ by differentiating it w.r.t. $w$ and setting it to 0:

$$\frac{\partial L(\mathbf{w}, \lambda)}{\partial \mathbf{w}} = \nabla \phi(\mathbf{w}) - \nabla \phi(\mathbf{w}_0) - \mathbf{X}^\top \lambda = 0. \tag{12}$$

Since $\nabla\phi^{-1}$ is the inverse of $\nabla\phi$, we obtain:

$$\mathbf{w}^\infty = \nabla\phi^{-1}\left(\nabla\phi(\mathbf{w}^0) + \mathbf{X}^\top\lambda\right), \tag{13}$$

where $\lambda$ will be chosen to satisfy $\mathbf{y} = \mathbf{X}\mathbf{w}^\infty$.

To obtain a closed-form solution, we can linearize Eq. (13) around $\nabla\phi(\mathbf{w}^0)$ using the fact that the mapping with the potential is invertible, so $\nabla\phi^{-1}\left(\nabla\phi(\mathbf{w}^0)\right) = \mathbf{w}^0$:

$$\mathbf{w}^\infty \approx \nabla\phi^{-1}\left(\nabla\phi(\mathbf{w}^0)\right) + \nabla^2\phi^{-1}\left(\nabla\phi(\mathbf{w}^0)\right)\mathbf{X}^\top\lambda = \mathbf{w}^0 + \mathbf{H}_{\phi^{-1}}\mathbf{X}^\top\lambda, \tag{14}$$

where we denoted $\mathbf{H}_{\phi^{-1}} = \nabla^2\phi^{-1}\left(\nabla\phi(\mathbf{w}^0)\right)$. For potentials that couple weights together, the Hessian will be non-diagonal. However, we use "local" potentials in which each entry $i$ of $\nabla\phi(\mathbf{w})$ depends only on $w_i$ (i.e. $\phi(\mathbf{w}) = \sum_i f(w_i)$ for some function $f$). For such potentials, the Hessian becomes diagonal. Since the mapping w.r.t. the potential is invertible, we can use the inverse function theorem to compute the Hessian $\mathbf{H}_{\phi^{-1}}$: $\mathbf{H}_{\phi^{-1}} = \nabla^2\phi^{-1}(\mathbf{z}) = \left(\nabla^2\phi(\mathbf{w}^0)\right)^{-1}$ for $\mathbf{z} = \nabla\phi(\mathbf{w}^0)$, assuming $\phi$ is twice continuously differentiable and $\mathbf{H}_{\phi^{-1}}$ is non-singular.

Since $\mathbf{H}_{\phi^{-1}}$ is ultimately a function of $\phi$, we have the structure of our solution. However, we need to solve for $\lambda$ in order to satisfy $\mathbf{y} = \mathbf{X}\mathbf{w}^\infty$. Assuming that $\mathbf{H}_{\phi^{-1}}$ is positive-definite and that $\mathbf{X} \in \mathbb{R}^{N\times D}$ is rank $N$ ($D \geq N$) for the sake of invertibility, $\lambda$ can be approximated by $\widehat{\lambda}$ as:

$$\lambda \approx \widehat{\lambda} = \left(\mathbf{X}\mathbf{H}_{\phi^{-1}}\mathbf{X}^\top\right)^{-1}\left(\mathbf{y} - \mathbf{X}\mathbf{w}^0\right). \tag{15}$$

With Eq. (15), we can now state the main result. Intuitively, what this result will show is that (1) we can know the shape of the distribution of the final weights, (2) that shape is independent of the loss function and the dataset, but not the initial weights. More formally, this is stated as:

**Theorem 1** (Informal). *Consider $N$ i.i.d. samples $y^n, \mathbf{x}^n$, such that: $\mathbf{x}^n \in \mathbb{R}^D$ are zero-mean and bounded; pairwise correlations $c_{ij} = \mathbb{E}\, x_i^n x_j^n$ and $c'_{ij} = \mathrm{Cov}((x_i^n)^2, (x_j^n)^2)$ between entries of a single $\mathbf{x}^n$ decay quickly enough so $\sum_{j=1}^\infty |c_{ij}| \leq const$ and $\sum_{j=1}^\infty |c'_{ij}| \leq const$ for all $i$; $y^n = (\mathbf{x}^n)^\top\mathbf{w}^*$; the teacher weights $\mathbf{w}^*$ and the initial weights $\mathbf{w}^0$ have zero-mean, $O(1/D)$ variance, i.i.d. entries, and finite 8th moment; $\nabla^2\phi^{-1}(w_i^0)$ has finite 1st and 2nd moments.*

*Then for $\widehat{\lambda} = \left(\mathbf{X}\mathbf{H}_{\phi^{-1}}\mathbf{X}^\top\right)^{-1}\left(\mathbf{y} - \mathbf{X}\mathbf{w}^0\right)$, individual entries of $\mathbf{X}^\top\widehat{\lambda}$ converge (in distribution) to a Gaussian with a constant variance $\sigma_\lambda^2$ as $D, N \to \infty$ with $N = o(D^{1/(5+\delta)})$ (for any $\delta > 0$):*

$$\frac{D\,h}{\sqrt{N}}\left(\mathbf{X}^\top\widehat{\lambda}\right)_i \longrightarrow_d \mathcal{N}(0, \sigma_\lambda^2), \tag{16}$$

*where $h = \mathbb{E}\left[\nabla^2\phi^{-1}(w_i^0)\right]$ (a scaling factor that is identical for all entries, which is not equivalent to $\mathbf{H}_{\phi^{-1}}$), and $\sigma_\lambda^2$ depends on the distributions of inputs and initial weights. The whole vector $\mathbf{X}^\top\lambda$ converges to a Gaussian process (over discrete indices) with weak correlations for distant points.*

*Proof sketch.* First, we show that scaled $\left(\mathbf{X}\mathbf{H}_{\phi^{-1}}\mathbf{X}^\top\right)^{-1}$ behaves like an identity for large $N, D$. Then, we show that $(\mathbf{X}^\top(\mathbf{y} - \mathbf{X}\mathbf{w}^0))_i$ is a sum of $N$ exchangeable variables (i.e. any permutation of these points has the same distribution) that satisfy conditions for a central limit theorem for such variables. Finally, we convert this result into a convergence to a Gaussian process. $\qquad\square$

We postpone the full proof, along with a version of the theorem for generic labels (without the teacher weights $\mathbf{w}^*$), to Appendix A. To unpack this theorem a bit more for the reader, combined with Eq. (13), the theorem shows that given some initial weights, $\mathbf{w}^0$, the distribution of $\mathbf{w}^\infty$ will depend on that initial distribution plus a term that converges to a Gaussian (in the dual space), as long as the loss has a unique finite root (defined before Eq. (8)) and the dataset correlations are small ($c_{ij}, c'_{ij}$ in Theorem 1). As such, the distribution of the solution weights will depend on the initial weights and the potential, but not on the loss or data. To clarify the assumptions used, we need the pairwise correlations $c_{ij}, c'_{ij}$ between distant inputs to be small. This is a common feature in natural sensory data Ruderman (1994) and neural activity Schulz et al. (2015). Finally, we note that the $N$ scaling with $D$ results from a generic large deviations bound and could likely be improved under some assumptions. In Section 4, we experimentally verify that $N = D^{0.5}$ and $N = D^{0.75}$ result in Gaussian behavior in the tested settings.

**Practical usage of our result.** Since we expect the weight change in the dual space to look Gaussian, we also expect the weight change in the dual space of a wrong potential to *not* be Gaussian. If we denote the true potential $\nabla\phi = f$ and another potential $\nabla\phi' = f'$, our result states that the change in the dual space is Gaussian: $f(\mathbf{w}^\infty) - f(\mathbf{w}^0) = \xi$, where $\xi$ is Gaussian. If we use the wrong potential, i.e. $f'$, we get $f'(\mathbf{w}^\infty) = f'(f^{-1}(f(\mathbf{w}^0) + \xi))$. If $f$ and $f'$ are similar, then we $f'(f^{-1}(\cdot))$ is approximately an identity, so we would see a (slightly less than for $f$) Gaussian change. If not, the change will be non-Gaussian due to the nonlinear $f'(f^{-1}(\cdot))$ transformation. Therefore we will be able to determine the potential used for training by finding the one with the most Gaussian change.

**Applicability to other learning rules.** Our result in Theorem 1 is made possible by two components: the structure of mirror descent solutions (see Eq. (8)) and the Gaussian behavior of large sums. However, the mirror descent framework can be applied to any learning rule, if we replace the gradient w.r.t. the loss in Eq. (5) with a generic error term $\mathbf{g}^t$: $\nabla\phi(\mathbf{w}^{t+1}) = \nabla\phi(\mathbf{w}^t) - \eta\,\mathbf{g}^t$. This way, for small and approximately independent (e.g. exchangeable) error terms $g^t$ their total contribution to the weight change would also follow the central limit theorem, resulting in a small Gaussian weight change. However, for generic error terms we cannot rely on Eq. (8) and can only provide generic conditions for Guassian behavior. Therefore, while our work leverages gradient-based optimization, the intuition we present applies more broadly.

**Theorem applicability in lazy and rich regimes.** The variance of the Gaussian term in Theorem 1 depends on the choice of the potential (via $h$). This means that the magnitude of the change from $\mathbf{w}^0$ to $\mathbf{w}^\infty$ will depend on $\phi$. Moreover, if the "learned" term $\mathbf{X}^\top\lambda$ in Eq. (13) is not small compared to $\nabla\phi(\mathbf{w}^0)$, our theory will not be applicable since the linearization will not be valid. Therefore, the applicability of our theorem is related to the question of "rich" versus "lazy" regimes of learning. Our theory is valid only in the lazy regime, in which learning is mostly linear Chizat et al. (2019).

However, whether we are in the rich or lazy regime turns out to depend on the potential. Assume for simplicity that all initial weights are the same and positive: $w_d^0 = w^0 = \alpha/\sqrt{D}$. The standard lazy regime corresponds to large weights with $\alpha = 1$, and the standard rich regime corresponds to small weights with $\alpha = 1/\sqrt{D}$. We can show (see Appendix A.1 for a derivation) that for $p$-norms, the dual weights are asymptotically larger than the weight changes as long as $\alpha \gg \sqrt{N/D}$ (for $N$ data points and $D$ weights). For negative entropy, this bound is more loose: $\alpha \gg \sqrt{N/(D\log D)}$. Therefore, for the same weight initialization and dataset size, negative entropy would typically produce smaller updates. For the standard for deep networks initialization with $\alpha \approx 1$, our theory should be applicable to datasets with $N \ll D$ (e.g. $N = D^{0.5}$).

## 4 EXPERIMENTS

Here we empirically verify our theory under conditions relevant for neuroscientific experiments. We use PyTorch Paszke et al. (2019) and FFCV library for fast data loading Leclerc et al. (2022). The experiments were performed on a local cluster with A100 NVIDIA GPUs. Experimental details are provided in Appendix B. Code is available at github.com/romanpogodin/synaptic-weight-distr.

### 4.1 LINEAR REGRESSION

We begin by testing whether the theorem holds in the case of linear regression, where we know that Eq. (8) holds. However, for experimental applicability, we want to verify that the histogram of $\xi_i$ over entries $i$ is Gaussian even when we draw it from a single network (rather than several initializations of a network), which would be the condition that holds in real neural experiments. As well, we want to verify that the change of weights $\xi_i = \nabla\phi(w_i^\infty) - \nabla\phi(w_i^0)$ is smaller than $\nabla\phi(w_i^0)$.

For each potential, we draw $x_i \sim \mathcal{N}(0,1) + \mathcal{U}(-0.5, 0.5)$ such that the correlation structure is as follows: $\mathbb{E}\,x_i\,x_j = (-1)^{|i-j|}/(1+|i-j|^c)$ for $c = 1, 2$. Note that $c = 1$ is a more relaxed condition than the correlation assumption used in proving Theorem 1. This is, therefore, a test of whether the theorem applies more broadly. In our experiments here, the initial weights are drawn from $w_i \sim \mathcal{N}(0, 1/D)$ for width $D$, and labels are drawn as $y = \pm 1$ (equal probability). We then optimize $(y - \sum_i x_i w_i)^2$ for networks of different widths $D$ and $N = D^{0.5}, D^{0.75}$. Strictly speaking only $N = o(D^{1/(5+\delta)})$ satisfies Theorem 1, so again, we are testing whether the theorem holds empirically when we relax our assumptions. We measure two things: the magnitude of weight changes in the

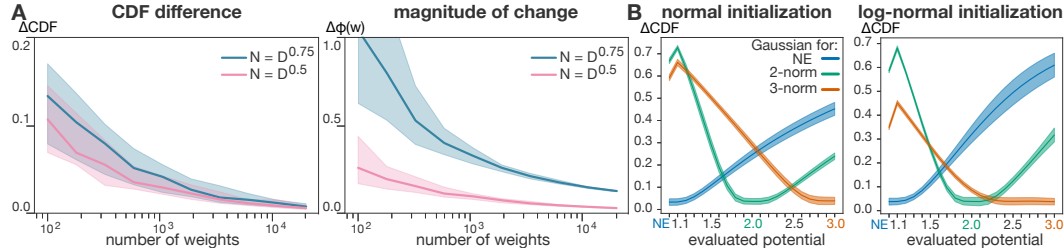

Figure 3: **A.** Linear regression solutions for negative entropy (NE) and fast correlation decay ($c = 2$). Left: integral of absolute CDF difference ($\Delta$CDF) between normalized uncentered weights and $\mathcal{N}(0, 1)$. Right: magnitude of weight changes relative to the initial weights in the dual space ($\Delta\phi$). Solid line: median over 30 seeds; shaded area: 5/95% percentiles; pink: $N = D^{0.5}$; blue: $N = D^{0.75}$. **B.** $\Delta$CDF for Gaussian (left) and log-normal (right) weight initializations and a Gaussian addition w.r.t. $\phi$, evaluated on another potential $\phi'$ (e.g. blue lines are sampled for NE but evaluated on every potential). Solid line: mean over 30 seeds; shaded areas: mean $\pm$ standard deviation.

dual space $\Delta\phi(\mathbf{w}) = \|\xi\|_2 / \|\nabla\phi(\mathbf{w}^0)\|_2$ (i.e. $\xi$ normalized relative to the initial weights) and $\Delta\text{CDF} = \int dt\, |\text{eCDF}(t) - \text{CDF}(t)|$ (i.e. the difference between the empirical cumulative density function, $\text{eCDF}(t)$ and the standard normal $\text{CDF}(t))$[1].

First, we find that $\Delta$CDF is always relatively small ($\leq 0.2$) and converges towards zero as the number of weights increases (Fig. 3A for negative entropy with $c = 2$. The full results are postponed to the appendix; see Fig. 6). Hence, we find that the weight changes in the dual space behave like a Gaussian when we know what potential was used for training, confirming Theorem 1. Second, we find that the magnitude of weight changes in the dual space $\Delta\phi(\mathbf{w})$ is much smaller than 1 for large widths, justifying the linearization in Eq. (14).

## 4.2 Robustness to potential change

We also need to test what happens if we don't know the true potential used for training, here denoted $\phi$, which is the case for neural data. As discussed after Theorem 1, for $\phi'$ close to the true potential we should see Gaussian changes, and for $\phi'$ far from $\phi$ – non-Gaussian ones. We test negative entropy and 2/3-norms as $\phi$, and $p$-norm with $p \in [1.1, 3]$ and negative entropy as $\phi'$; $\mathbf{w}^0$ is drawn from either Gaussian or log-normal distributions with approximately the same variance. We draw a Gaussian $\xi_\phi$ to compute $\mathbf{w}^\infty$ using $\phi$ from $\xi_\phi = \nabla\phi(\mathbf{w}^\infty) - \nabla\phi(\mathbf{w}^0)$. We find that the "empirical" change for $\phi'$, $\xi_{\phi'} = \nabla\phi'(\mathbf{w}^\infty) - \nabla'\phi(\mathbf{w}^0)$, is indeed Gaussian only for small variations from the true potential $\phi$ (Fig. 3B). In particular, the distinction between negative entropy and other, non-multiplicative updates, is very pronounced.[2] For $\phi = 3$-norm and log-normal initialization, the range for a Gaussian $\xi_{\phi'}$ almost reached $p = 2.3$ (Fig. 3B, bottom, orange line). Thus, if we hypothesize a potential, $\phi'$, that is similar to the true potential $\phi$, we get a nearly Gaussian $\xi_{\phi'}$. In contrast, if we have the wrong form for the potential (e.g. the potential is the negative entropy but we hypothesize a 3-norm potential), then we get a clearly non-Gaussian distribution. As such, we can use the distribution of $\xi_{\phi'}$ to test hypotheses about synaptic geometry.

## 4.3 Finetuning of deep networks

We expect our theory to work for networks that behave similarly to their linearization during learning. One such example is a pretrained trained network that's being finetuned on new data. If the new data comes from a similar distribution as the pretraining data, then the linearized network should approximate its non-linearized counterpart well with respect to weight changes Mohamadi et al. (2022); Ren et al. (2023). This is also the scenario we expect to see in neural data if we are training an animal on an ethologically relevant task that matches its prior life experience and innate capabilities to a reasonable degree.

---

[1] We use normalized but uncentered weight distributions, since we predict the change $\xi$ to be zero-mean.

[2] Negative entropy is placed at $p = 1$, as for weights that sum to 1, $p$−norms are equivalent to the shifted and scaled negative Tsallis entropy, which tends to negative entropy as $p \to 1$.

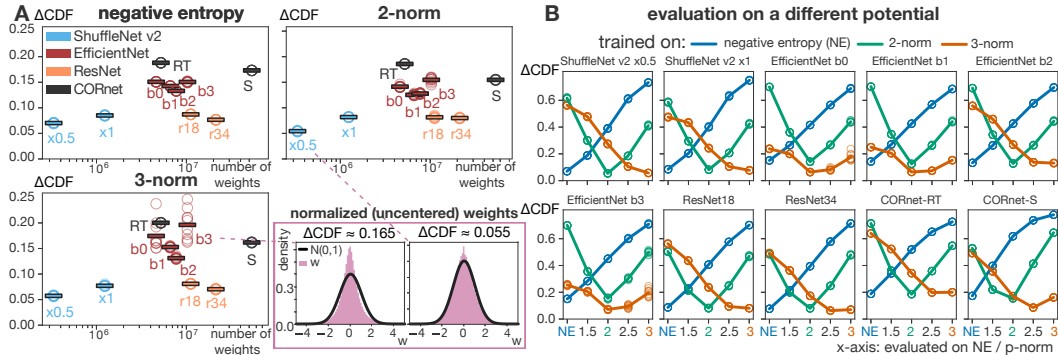

Figure 4: Finetuning on 10 randomly sampled ImageNet validation subsets ($N = D^{0.5}$ data points). **A.** Integral of absolute CDF difference ($\Delta$CDF) between normalized uncentered weights and $\mathcal{N}(0,1)$ for networks trained with different potentials. Circle: individual value; bar: mean over seeds. Pink box (bottom right): examples of weight change histograms (pink) plotted against $\mathcal{N}(0,1)$ (black). **B.** Same as **A.**, but $\Delta$CDF is calculated w.r.t. other potentials.

Finetuning deep networks still breaks several assumption of Theorem 1: the initial weights are not i.i.d., the network is not linearized (although potentially close to it), and the data correlations are unknown. Moreover, the cross-entropy loss we use is technically not a loss with a unique finite root. However, the readout layers in the deep networks that we test have fewer units than the number of ImageNet Deng et al. (2009) classes, so the activity in the readout layer is not linearly separable and the weights do not tend to infinity. Yet, they are still large and change significantly during training, so we exclude the readout layer when assessing weight distributions (along with biases and BatchNorm parameters). We use networks pretrained on ImageNet, and finetune them to 100% accuracy on a subset of ImageNet validation set. The pretrained networks have not seen the validation data, but they already perform reasonably well on it, so there's no distribution shift in the data that would force the networks to change their weights a lot. We used $N = D^{0.5}$ data points for $D$ weights ($N = D^{0.75}$ exceeded dataset size) and four architecture types: ShuffleNet v2 Ma et al. (2018) (x0.5/x1), EfficientNet Tan & Le (2019) (b0-b3), ResNet He et al. (2016) (18/34), and CORnet Kubilius et al. (2019) (S/RT) chosen to span a wide range of parameter counts (0.3M to 53.4M). CORnets were also chosen since they mimic the primate ventral stream and its recurrent connections.

For all three tested potentials, weight changes are close to a Gaussian distribution ($\Delta$CDF $< 0.2$; Fig. 4A). ShuffleNet v2s and ResNets reach consistently more Gaussian solutions than EfficientNets and CORnets, despite ShuffleNet v2s having fewer parameters. This suggests that some trained architectures are more "linear", which may be due to the architecture itself, or the way the networks were trained. The magnitude of changes in the dual space was typically smaller than 0.2 (see Appendix B). Just like for the toy task in Fig. 3B, if we trained with a specific potential, $\phi$, we only observed Gaussian changes when the hypothesized potential, $\phi'$, was close to $\phi$ (Fig. 4B). The only exceptions were 3-norm-trained EfficientNets 0,1,3 and CORnet-S, for which the 2-norm solution was slightly more Gaussian than the 3-norm solution even when $\phi$ was a 3-norm. However, this difference was small, and the networks had overall worse fit than other architectures. In all cases, using negative entropy as $\phi'$ provided the best fit for negative entropy-trained networks and the worst fit for other potentials. This is important because negative entropy results in multiplicative weights updates, while $p$-norms result in additive updates, which represent very distinct hypotheses for synaptic plasticity mechanisms. Taken together, these empirical results suggest that Theorem 1 holds more generally and can be used to infer the distribution of weight changes beyond linear regression.

## 4.4 ESTIMATING SYNAPTIC GEOMETRY EXPERIMENTALLY

Previously we showed that Theorem 1 holds when finetuning pretrained deep networks, despite non-i.i.d. initial weights and an unknown input correlation structure. As such, these finetuning experiments indicate our theory is applicable to neuroscience experiments when we measure the distribution of weights before and after learning, and use the histogram of weight changes to estimate the synaptic geometry. In this section we apply this technique to experimental neuroscience data.

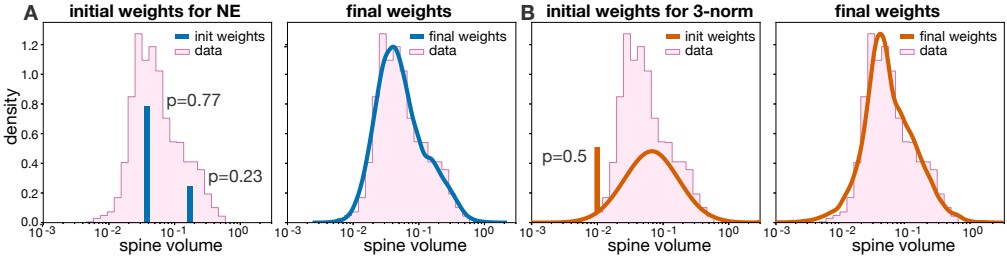

Figure 5: Observed vs. modeled synaptic distributions. Pink: spine volume ($\mu m^3$) from Dorkenwald et al. (2022); vertical bars: point mass initializations. **A.** Left: initial weights for negative entropy (NE); right: final weight for a Gaussian change in the dual space. **B.** Same as **A**, but for 3-norm.

Specifically, we find that if one knows the initial weights, $\mathbf{w}^0$, then it is possible to distinguish between different potentials. But if the initial weights are unknown, then multiple potential functions can fit the data. To show this, we use data from a recent experimental study that measured an analogue of synaptic weights (synaptic spine size) using electron microscopy Dorkenwald et al. (2022) (pink histogram in Fig. 5). In the log space, Dorkenwald et al. (2022) found that the distribution was well modelled by a mixture of two Gaussians with approximately the same variance (pink histogram in Fig. 5) – instead of one Gaussian as has been previously reported Loewenstein et al. (2011). First, we show that the experimental data is consistent with the negative entropy potential (Fig. 5A), when using the following initialization: consider elements of $\mathbf{w}^0$ equal to a constant $\mu_1$ with probability $p$, and to $\mu_2$ with probability $1-p$. Then, $\nabla\phi(\mathbf{w}^\infty) = \nabla\phi(\mathbf{w}^0) + \xi$ will be a mixture of two Gaussians with the same variance but different means, and Fig. 5A shows the $\xi$ that fits the experimental data (parameters from Dorkenwald et al. (2022) with equalized variance, see Appendix B.4). However, the data is also consistent with the 3-norm potential for a different initialization: if $\mathbf{w}^0$ is a mixture of a constant and a log-normal, then adding an appropriate $\xi$ in the dual space also results in a weight distribution that resemble a mixture of two log-normals and appears to fit the experimental data well (Fig. 5B). Thus, if our goal is to determine the synaptic geometry of the brain, then it is important to estimate the synaptic weight distributions before and after training. Nevertheless, with this data we can rule out a Euclidean synaptic geometry, and if we do have access to $\mathbf{w}^0$, then our results show that it is indeed possible to experimentally estimate the potential function.

## 5    DISCUSSION

We presented a mirror descent-based theory of what synaptic weight distributions can tell us about the underlying synaptic geometry of a network. For a range of loss functions and under mild assumptions on the data, we showed that weight distributions depend on the synaptic geometry, but not on the loss or the training data. Experimentally, we showed that our theory applies to finetuned deep networks across different architectures (including recurrent ones) and network sizes. Thus, our theory would likely apply as well to the hierarchical, recurrent architectures seen in the brain. Our work predicts that if we know synaptic weights before and after learning, we can find the underlying synaptic geometry by finding a transformation in which the weight change is Gaussian.

It is important to note that by adopting the mirror descent framework we made the assumption that the brain would seek to achieve the best performance increases for the least amount of synaptic change possible. But, these results could be extended to learning algorithms that are not explicitly derived from that principle, such as three-factor Hebbian learning Frémaux & Gerstner (2016); Kuśmierz et al. (2017); Pogodin & Latham (2020). For a single layer, three-factor updates between $y$ and $\mathbf{x}$ follow $\epsilon\,y\,\mathbf{x}$ for some error signal $\epsilon$. As long as $\epsilon$ dynamics leads to a solution, the analysis should be similar to the general mirror descent one since weights will span $\mathbf{x}$ (as required by the theory). For multi-layered networks, all we would need to assume is that the input to each layer does not change too much over time, similar to the linearization argument made for deep networks.

More broadly, our work opens new experimental avenues for understanding synaptic plasticity in the brain. Our approach isolates synaptic geometry from other components of learning, such as losses, error signals, error pathways, etc. Therefore, our findings make it practical to experimentally determine the brain's synaptic geometry.

## ACKNOWLEDGMENTS

This work was supported by the following sources. BAR: NSERC (Discovery Grant RGPIN-2020-05105, RGPIN-2018-04821; Discovery Accelerator Supplement: RGPAS-2020-00031; Arthur B. McDonald Fellowship: 566355-2022), Healthy Brains, Healthy Lives (New Investigator Award: 2b-NISU-8), and CIFAR (Canada AI Chair; Learning in Machine and Brains Fellowship). GL: NSERC (Discovery Grant RGPIN-2018-04821), Canada-CIFAR AI Chair, Canada Research Chair in Neural Computations and Interfacing, and Samsung Electronics Co., Ldt. GG: Canada CIFAR AI Chair. JC: IVADO Postdoctoral Fellowship and the Canada First Research Excellence Fund. AG: Vanier Canada Graduate scholarship.

This research was enabled in part by support provided by (Calcul Québec) (`https://www.calculquebec.ca/en/`) and Compute Canada (`www.computecanada.ca`). The authors acknowledge the material support of NVIDIA in the form of computational resources.

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

APPENDICES

## A  WEIGHT DISTRIBUTION CONVERGENCE

We start by giving an intuitive outline of our proof of Gaussian convergence of weight changes. As for any Gaussian convergence, we'll end up using the central limit theorem (although an unusual version of it) w.r.t. the dataset size. With this in mind, the steps are:

Setup  First, we decide how we take the large data and network limits. We have a vector of weight changes whose size grows to infinity. To discuss what it converges to, we consider this vector to be a part of a stochastic process (so, an already infinite-dimensional vector). We then consider the convergence of this stochastic process to some limiting process (in our case, to a Gaussian process). This allows us to approximate a sample of weight changes as a sample from that limiting process.

Lemma 1  Next, we show that the inverse matrix in Eq. (15) used in the weight change expression behaves like an identity matrix. If we just took the limit w.r.t. the network size, this lemma would be trivial since off-diagonal elements are products of independent vectors. But we also increase the dataset size (and hence the matrix itself), so we need to be more careful. This is where we get (most of) the requirements for how the dataset size should scale with network size.

Lemma 2  shows that the deviations from identity resulting from Lemma 1 don't affect the final weight change expression when we take the limits.

Lemma 3  shows that if you replace that inverse matrix with an identity, you can apply the exchangeable central limit theorem to the weight change expression. At this point, it's applied for any finite slice of the stochastic process we defined. The exchangeable central limit theorem has much more restrictive moment conditions than the standard CLT, since it doesn't require independence, so we have to carefully check all of them.

Theorem 1  Finally, we combine the lemmas and show that any finite slice of that stochastic process converges to a Gaussian random variable. Due to the properties of Gaussian processes, we can conclude that the whole process converges to a Gaussian process.

**Setup**  We want to show that the weight change looks Gaussian in the large width limit. Since the number of weights goes to infinity, "looks Gaussian" should translate to convergence to a Gaussian process. Our approach will be similar to Matthews et al. (2018), which showed convergence of wide networks to Gaussian processes. However, they worked with finite-dimensional inputs and multiple hidden layers; for our purposes we'll need to have an infinite-dimensional input and a single layer.

We'll be working with countable stochastic processes defined over $\mathbb{Z}_{>0}$. We need three stochastic processes: the input data $X$, the initial weights $W^0$ and the "teacher" weights $W^*$.

For width $D$, the label $y_D$ and the model at initialization $\widehat{y}_D^0$ are defined as

$$y_D = \sum_{d=1}^{D} \frac{1}{\sqrt{D}} X_d W_d^*, \quad \widehat{y}_D^0 = \sum_{d=1}^{D} \frac{1}{\sqrt{D}} X_d W_d^0. \tag{17}$$

For $N$ sample points of dimension $D$, we stack them into an $N \times D$ matrix $\mathbf{X}^D$, and denote the $D$-dimensional samples of weights as $\mathbf{w}^{*D}$, $\mathbf{w}^{0D}$. The weight change we're interested in is then

$$\xi^D = \frac{1}{\sqrt{D}} \left(\mathbf{X}^D\right)^\top \left(\mathbf{X}^D \mathbf{H}^D \left(\mathbf{X}^D\right)^\top\right)^{-1} \mathbf{X}^D (\mathbf{w}^{*D} - \mathbf{w}^{0D}) \tag{18}$$

$$= \left(\mathbf{X}^D\right)^\top \left(\mathbf{X}^D \mathbf{H}^D \left(\mathbf{X}^D\right)^\top\right)^{-1} (\mathbf{y}_D - \mathbf{y}_D^0). \tag{19}$$

Note the explicit $1/\sqrt{D}$ scaling we omitted in the main text for convenience and $N$-dimensional vectors of labels $\mathbf{y}_D$, $\mathbf{y}_D^0$. Here $\mathbf{H}^D$ is a diagonal matrix with $h_{dd} = \nabla^2 \phi^{-1}(W_d^0/\sqrt{D})$ (we dropped the $\phi^{-1}$ subscript used in the main text for readability).

**Invertibility assumption.** Throughout the proofs we assume that $\mathbf{X}^D \mathbf{H}^D \left(\mathbf{X}^D\right)^\top$ is almost surely invertible. This is a very mild assumption, although it can break if $X_d^i$ are sampled from a discrete

distribution, if $X_d$ have strong correlations along $d$ or if the initial weights are *not* almost surely non-zero.

$\xi^D$ is a $D$-dimensional vector, but we understand it as a subset of the following stochastic process:

$$\Xi^D = (\mathbf{X})^\top \left( \mathbf{X}^D \, \mathbf{H}^D \, (\mathbf{X}^D)^\top \right)^{-1} (\mathbf{y}_D - \mathbf{y}_D^0). \tag{20}$$

This infinite-width expansion is similar to the one used in Matthews et al. (2018). It's a well-defined stochastic process since $N$ is finite, meaning that $\Xi^D$ is a weighted sum of $N$ stochastic processes. However, it formalizes convergence of a finite-dimensional vector $\xi^D$ to something infinite-dimensional. That is, we'll work with convergence of the process $\Xi^D$ to a limit process $\Xi$ in distribution.[3]

First, we show that an appropriately scaled matrix inside the inverse behaves like an identity:

**Lemma 1.** *Assume that the input data points $X_d$ have zero mean and unit variance. Additionally assume that uniformly for any d, there exists a constant c such that*

$$\sum_{d'=1}^{\infty} (\mathbb{E} \, X_d \, X_{d'})^2 \le c, \quad \sum_{d'=1}^{\infty} \left| \text{Cov} \left( X_d^2, X_{d'}^2 \right) \right| \le c. \tag{21}$$

*Also for i.i.d. initial weights, define moments (assuming they exist) of $h_{dd} = \phi^{-1}(W_d^0/\sqrt{D})$ as*

$$h_1(D) = \mathbb{E} \, \nabla^2 \phi^{-1}(W_d^0/\sqrt{D}), \quad h_2(D) = \text{Var} \left( \nabla^2 \phi^{-1}(W_d^0/\sqrt{D}) \right). \tag{22}$$

*Then for $N, D$ large enough (so the r.h.s. is smaller than 4),*

$$\left\| \left( \frac{1}{D \, h_1(D)} \mathbf{X}^D \, \mathbf{H}^D \, (\mathbf{X}^D)^\top \right)^{-1} - \mathbf{I}_N \right\|_2 \le 8 \sqrt{\frac{N^4}{D} \frac{c+1}{p} \left( 1 + \frac{h_2}{h_1^2} \right)} \tag{23}$$

*with probability at least $1 - p$.*

*Proof.* Denote

$$\mathbf{A} \equiv \frac{1}{D \, h_1} \mathbf{X}^D \, \mathbf{H}^D \, (\mathbf{X}^D)^\top, \quad A_{ij} = \frac{1}{D \, h_1} \sum_{d=1}^{D} X_d^i \, X_d^j \, h_{dd}, \tag{24}$$

dropping the explicit $D$-dependence in $h_1$ and $h_2$ for convenience.

Since all $X_d$ are independent from $W_d^0$, $\mathbb{E} \, A_{ij} = \delta_{ij}$. Variance can be bounded. For $i = j$,

$$\text{Var}(A_{ii}) = \frac{1}{D^2 \, h_1^2} \sum_{d=1}^{D} \text{Var}((X_d^i)^2 \, h_{dd}) + \frac{1}{D^2 \, h_1^2} \sum_{d,d' \neq d}^{D} \text{Cov}((X_d^i)^2 \, h_{dd}, \, (X_{d'}^i)^2 \, h_{d'd'}) \tag{25}$$

$$= \sum_{d=1}^{D} \frac{h_2 + (h_1^2 + h_2) \, \text{Var}((X_d^i)^2)}{D^2 \, h_1^2} + \sum_{d,d' \neq d}^{D} \frac{h_1^2 \, \text{Cov} \left( (X_d^i)^2, (X_{d'}^i)^2 \right)}{D^2 \, h_1^2} \tag{26}$$

$$\le \frac{h_2(1 + \text{Var}((X_d^i)^2))}{D \, h_1^2} + \frac{c}{D} \le \frac{c}{D} \left( 1 + \frac{h_2}{h_1^2} \right) + \frac{h_2}{D \, h_1^2} \le \frac{c+1}{D} \left( 1 + \frac{h_2}{h_1^2} \right), \tag{27}$$

where in the last line we used the second part of Eq. (21).

For $i \neq j$,

$$\text{Var}(A_{ij}) = \frac{1}{D^2 \, h_1^2} \sum_{d=1}^{D} \text{Var}(X_d^i X_d^j \, h_{dd}) + \frac{1}{D^2 \, h_1^2} \sum_{d,d' \neq d}^{D} \text{Cov}(X_d^i X_d^j \, h_{dd}, \, X_{d'}^i X_{d'}^j \, h_{d'd'}) \tag{28}$$

$$= \sum_{d=1}^{D} \frac{(h_1^2 + h_2) \left( \text{Var}(X_d^i) \right)^2}{D^2 \, h_1^2} + \sum_{d,d' \neq d}^{D} \frac{h_1^2 \left( \text{Cov}(X_d^i, X_{d'}^i) \right)^2}{D^2 \, h_1^2} \le \frac{c}{D} \left( 1 + \frac{h_2}{h_1^2} \right), \tag{29}$$

---

[3]See Eq. 10 in Matthews et al. (2018) for the discussion on what "in distribution" means for stochastic processes.

now using the first part of Eq. (21) for the last inequality.

Using Chebyshev's inequality and the union bound, we can bound

$$\mathbb{P}\left(\max_{ij}|A_{ij}-\delta_{ij}|\geq\epsilon\right)\leq\frac{N^2}{\epsilon^2}\frac{c+1}{D}\left(1+\frac{h_2}{h_1^2}\right)\equiv p\,. \tag{30}$$

Therefore,

$$\|\mathbf{A}-\mathbf{I}_N\|_2\leq\|\mathbf{A}-\mathbf{I}_N\|_F\leq\epsilon\,N=\sqrt{\frac{N^4}{D}\frac{c+1}{p}\left(1+\frac{h_2}{h_1^2}\right)} \tag{31}$$

with probability at least $1-p$.

By Lemma 4.1.5 of Vershynin (2018), if $\|\mathbf{A}-\mathbf{I}_N\|_2\leq\epsilon\,N$ then all singular values $\sigma_i$ of $\mathbf{A}$ lie between $(1-\epsilon\,N)^2$ and $(1+\epsilon\,N)^2$ for $N$ large enough so $\epsilon\,N<1$. Therefore, we also have a bound on the singular values of $\mathbf{A}^{-1}$. Additionally taking parameters big enough for $\epsilon\,N\leq 0.5$, we have

$$\|\mathbf{A}^{-1}-\mathbf{I}_N\|_2=\max_i\left(\frac{1}{\sigma_i}-1\right)\leq\frac{1}{(1-\epsilon\,N)^2}-1=\frac{2\epsilon\,N-\epsilon^2 N^2}{(1-\epsilon\,N)^2}\leq 8\,\epsilon\,N\,, \tag{32}$$

which completes the proof. $\qquad\square$

Now we need to show that the inverse behaves like an identity in the overall expression as well:

**Lemma 2.** *In the assumptions of Lemma 1, additionally assume that $W_d^*$ and $W_d^0$ have finite mean and variance independent of $d$ and $D$, that $h_2(D)/h_1(D)^2$ is independent of $D$, and $\sum_{d'=1}^{\infty}\mathbb{E}\,X_d\,X_{d'}\leq c$ uniformly over $d$. Then for any scalar weights $\alpha_i$ and a finite index set $\mathcal{A}\subset\mathbb{Z}_{>0}$, as $D,\,N\to\infty$ with $N=o(D^{1/(5+\delta)})$ for any $\delta>0$,*

$$\frac{1}{\sqrt{N}}\sum_{p\in\mathcal{A}}\alpha_p\,(\mathbf{X}_{:p})^{\top}\left(\mathbf{A}^{-1}-\mathbf{I}_N\right)(\mathbf{y}_D-\mathbf{y}_D^0)\to_p 0\,. \tag{33}$$

**Remark.** For both cross-entropy and $p$-norm potentials, $\nabla^2\phi^{-1}(W/\sqrt{D})$ is a power of $W/\sqrt{D}$, and so $h_2(D)/h_1(D)^2$ depends only on the moments of $W$, but not on $D$, therefore satisfying the assumption.

*Proof.* Again denoting the matrix inside the inverse as $\mathbf{A}$ (see Eq. (24)), we first bound the following via Cauchy-Schwarz:

$$\left|\sum_{i\in\mathcal{A}}\alpha_i\,(\mathbf{X}_{:i})^{\top}\left(\mathbf{A}^{-1}-\mathbf{I}_N\right)(\mathbf{y}_D-\mathbf{y}_D^0)\right|^2\leq\left\|\sum_{i\in\mathcal{A}}\alpha_i\,(\mathbf{X}_{:i})\right\|_2^2\left\|\mathbf{y}_D-\mathbf{y}_D^0\right\|_2^2\left\|\mathbf{A}^{-1}-\mathbf{I}_N\right\|_2^2\,. \tag{34}$$

**First term.** Using Cauchy-Schwarz and that $X_p$ variances are one, and also that $\alpha$, $\mathcal{A}$ are fixed,

$$\mathbb{E}\left\|\sum_{p\in\mathcal{A}}\alpha_p\,(\mathbf{X}_{:p})\right\|_2^2=N\,\mathbb{E}\left(\sum_{p\in\mathcal{A}}\alpha_p\,X_p^1\right)^2\leq N\,\mathbb{E}\left(\sum_{p\in\mathcal{A}}\alpha_p^2\right)\left(\sum_{p\in\mathcal{A}}(X_p^1)^2\right)=O(N)\,. \tag{35}$$

**Second term.** Using that weights and inputs are independent, and that the sum of input data covariances is bounded by $c$ by our assumption,

$$\mathbb{E} \left\| \mathbf{y}_D - \mathbf{y}_D^0 \right\|_2^2 = \frac{N}{D} \mathbb{E} \left( \sum_{d=1}^{D} X_d^1 \left( W_d^* - W_d^0 \right) \right)^2 \tag{36}$$

$$= \frac{N}{D} \sum_{d,d' \neq d}^{D} \left( \mathbb{E} \left( W_d^* - W_d^0 \right) \right)^2 \mathbb{E} \left( X_d^1 X_{d'}^1 \right) \tag{37}$$

$$+ \frac{N}{D} \sum_{d=1}^{D} \left( \left( \mathbb{E} \left( W_d^* - W_d^0 \right) \right)^2 + \mathbb{V}\mathrm{ar}\left( \left( W_d^* - W_d^0 \right) \right) \right) \mathbb{E} \left( X_d^1 \right)^2 \tag{38}$$

$$\leq N\, c \left( \mathbb{E} \left( W_d^* - W_d^0 \right) \right)^2 + N\, \mathbb{V}\mathrm{ar}\left( \left( W_d^* - W_d^0 \right) \right) = O(N)\,. \tag{39}$$

**Third term.** Here we will use the probabilistic bound from Lemma 1.

Overall, for any $\delta > 0$,

$$\mathbb{P} \left( \frac{1}{N} \left| \sum_{i \in \mathcal{A}} \alpha_i \left( \mathbf{X}_{:\,i} \right)^\top \left( \mathbf{A}^{-1} - \mathbf{I}_N \right) \left( \mathbf{y}_D - \mathbf{y}_D^0 \right) \right|^2 > \epsilon \right) \tag{40}$$

$$\leq \mathbb{P} \left( \left\| \sum_{i \in \mathcal{A}} \alpha_i \left( \mathbf{X}_{:\,i} \right) \right\|_2^2 > N^{1+\frac{\delta}{2}};\, \left\| \mathbf{y}_D - \mathbf{y}_D^0 \right\|_2^2 > N^{1+\frac{\delta}{2}};\, \left\| \mathbf{A}^{-1} - \mathbf{I}_N \right\|_2^2 > \frac{\epsilon}{N^{1+\delta}} \right) \tag{41}$$

$$\leq \frac{\mathbb{E} \left\| \sum_{p \in \mathcal{A}} \alpha_p \left( \mathbf{X}_{:\,p} \right) \right\|_2^2}{N^{1+\frac{\delta}{2}}} + \frac{\mathbb{E} \left\| \mathbf{y}_D - \mathbf{y}_D^0 \right\|_2^2}{N^{1+\frac{\delta}{2}}} + \mathbb{P} \left( \left\| \mathbf{A}^{-1} - \mathbf{I}_N \right\|_2^2 > \frac{\epsilon}{N^{1+\delta}} \right) \tag{42}$$

$$\leq O \left( \frac{1}{N^{\delta/2}} \right) + 64 \frac{N^{5+\delta}}{D} \frac{c+1}{\epsilon} \left( 1 + \frac{h_2}{h_1^2} \right) \tag{43}$$

where we used Eq. (34) in the first line, the union bound with Markov's inequality in the second and the bounds on all terms in the third one (for $N$ large enough so $\epsilon/N^{1+\delta} < 2$).

As long as $N^{5+\delta} = o(D)$ and $h_2/h_1^2$ is constant, the probability goes to zero with $N, D$. $\qquad\square$

The previous result suggests we can replace the inverse matrix in $\Xi^D$ (Eq. (20)) with an identity. This will require an additional step, but now we can show that with the identity, for any finite index set $\mathcal{A}$ of points in $\Xi^D$ and arbitrary weights $\alpha$, the $\mathcal{A}$, $\alpha$-projection of $\Xi^D$ solution converges to a Gaussian.

**Lemma 3.** *Consider a finite index set $\mathcal{A}$ and an associated vector of weights $\alpha$. If in addition to condition of Lemmas 1 and 2 we assume weights have a finite 8th moment, $X_p^i$ are uniformly bounded, and that $X_p^i(\mathbf{y}_D - \mathbf{y}_D^0)_i$ (for point $i$ and index $p$) converges in probability to a random variable. Then the $\mathcal{A}$, $\alpha$-projection defined as,*

$$\frac{1}{\sqrt{N}} \sum_{i=1}^{N} R_{Ni}, \quad R_{Ni} = \sum_{p \in \mathcal{A}} \alpha_p X_p^i(\mathbf{y}_D - \mathbf{y}_D^0)_i\,, \tag{44}$$

*converges to $\mathcal{N}(0, \sigma_W^2\, \alpha^\top \Sigma_{\mathcal{A}} \alpha)$ in distribution, where $(\Sigma_{\mathcal{A}})_{pp'} = \mathbb{E}\, X_p^i X_{p'}^i$ and $\sigma_W^2 = \mathbb{E}\,(W_d^* - W_d^0)^2$, as long as $D$ increases monotonically with $N$ such that $N/D = o(1)$.*

*Proof.* We assumed that $D$ is a monotonically increasing function of $N$. Therefore, we get a triangular array of $R_{Ni}$ with infinitely exchangeable rows. That is, for each $N$ we have a process $R_{Ni}$ with $i = 1, \ldots$, such that any permutation of indices $i$ doesn't change the joint distribution. This is due to joint weight dependence among labels. Sums of infinitely exchangeable sequences behave similarly to sums of independent sequences when it comes to limiting distributions. In particular, we will use the classic central limit theorem result of Blum et al. (1958).

To apply it, we need to compute several moments of $R_{Ni}$.

**First moment.** Since $\mathbb{E}(W_d^* - W_d^0) = 0$, we have $\mathbb{E}\, R_{Ni} = 0$.

**Second moment.** Again using $\mathbb{E}\,(W_d^* - W_d^0) = 0$ and the fact that all weights are i.i.d., so $\mathbb{E}\,(W_d^* - W_d^0)(W_{d'}^* - W_{d'}^0) = \delta_{ij}$, we get

$$\sigma_R^2 = \mathbb{E}\, R_{Ni}^2 = \frac{1}{D} \sum_{pp'dd'} \alpha_p\, \alpha_{p'}\, \mathbb{E}\, X_p^i X_{p'}^i X_d^i X_{d'}^i (W_d^* - W_d^0)(W_{d'}^* - W_{d'}^0) \tag{45}$$

$$= \frac{1}{D} \sum_{pp'd} \alpha_p\, \alpha_{p'}\, \mathbb{E}\, X_p^i X_{p'}^i (X_d^i)^2 (W_d^* - W_d^0)^2\,. \tag{46}$$

This term is $O(1)$ due to the $1/D$ scaling. Denoting $\sigma_W^2 = \mathbb{E}\,(W_d^* - W_d^0)^2$ and $\gamma_{pp'}^D = \sum_d \mathbb{E}\, X_p^i X_{p'}^i (X_d^i)^2 / D$ (and the corresponding matrix $\Gamma^D$), we can re-write the variance as

$$\sigma_R^2(D) = \sigma_W^2\, \alpha^\top \Gamma^D \alpha\,. \tag{47}$$

**Absolute 3rd and 4th moments.** For convenience, we first find the 4th moment. Since the inputs are bounded and independent from the weights, and dropping the summation over all $p_i$ and $d_i$ for readability,

$$\mathbb{E}\, R_{Nj}^4 = \frac{1}{D^2} \sum \left( \prod_{i=1}^4 \alpha_{p_i} \right) \mathbb{E}\left( \prod_{i=1}^4 X_{p_i}^j X_{d_i}^j \right) \mathbb{E}\left( \prod_{i=1}^4 (W_{d_i}^* - W_{d_i}^0) \right) = O(1)\,. \tag{48}$$

This follows from $\mathbb{E}\left( \prod_{i=1}^4 X_{p_i}^j X_{d_i}^j \right) = O(1)$ due to bounded inputs, and due to centered and i.i.d. weights $\mathbb{E}\left( \prod_{i=1}^4 (W_{d_i}^* - W_{d_i}^0) \right) = 0$ if any $d_i$ differs from the rest (i.e. only $O(D^2)$ out of $D^4$ are non-zero; these are the ones with $d_1 = d_2$ *and* $d_3 = d_4$ and permutations of those conditions).

For the absolute 3rd moment, we can use Cauchy-Schwarz to find its scaling:

$$\mathbb{E}\, |R_{Nj}|^3 = \mathbb{E}\, |R_{Nj}| \cdot |R_{Nj}|^2 \le \sqrt{\mathbb{E}\, |R_{Nj}|^2 \, \mathbb{E}\, |R_{Nj}|^4} = O(1)\,, \tag{49}$$

since both terms in the square root are $O(1)$.

**Covariance.** For two different points $i$ and $j$, defining the covariance $\sigma_{pd} = \mathbb{E}\, X_p^i X_d^i$,

$$\mathbb{E}\, R_{Ni} R_{Nj} = \frac{1}{D} \sum_{pp'dd'} \alpha_p\, \alpha_{p'}\, \mathbb{E}\, X_p^i X_{p'}^j X_d^i X_{d'}^j (W_d^* - W_d^0)(W_{d'}^* - W_{d'}^0) \tag{50}$$

$$= \frac{1}{D} \sum_{pp'd} \alpha_p\, \alpha_{p'}\, \mathbb{E}\, X_p^i X_{p'}^j X_d^i X_d^j (W_d^* - W_d^0)^2 = \frac{\sigma_W^2}{D} \sum_{pp'd} \alpha_p \alpha_{p'} \sigma_{pd} \sigma_{p'd}\,. \tag{51}$$

Since we required $\sum_d \sigma_{pd}^2 \le c$, $\sum_d \sigma_{pd} \sigma_{p'd} \le \sqrt{(\sum_d \sigma_{pd}^2)(\sum_d \sigma_{p'd}^2)} \le c$ by Cauchy-Schwarz. Thus, $\mathbb{E}\, R_{Ni} R_{Nj} = O(1/D) = o(1/N)$.

**Covariance of $R_{Ni}^2$.** Again, take $i \ne j$ and adapt the calculation of the 4th moment:

$$\mathbb{E}\, R_{Ni}^2 R_{Nj}^2 = \frac{1}{D^2} \sum \left( \prod_{k=1}^4 \alpha_{p_k} \right) \mathbb{E}\left( \prod_{k=1}^2 X_{p_k}^i X_{d_k}^i \right) \left( \prod_{k=3}^4 X_{p_k}^j X_{d_k}^j \right) \left( \prod_{k=1}^4 (W_{d_k}^* - W_{d_k}^0) \right)\,. \tag{52}$$

Due to i.i.d. weights the whole sum has only $O(D^2)$ terms and therefore the whole expression is $O(1)$. However, we require a more precise control over the sum.

The summands split up into three cases (with a potential $O(D)$ overlap): (1) $d_1 = d_2 = d_3 = d_4$, (2) $d_1 = d_3,\ d_2 = d_4$ or $d_1 = d_4,\ d_2 = d_3$, and (3) $d_1 = d_2,\ d_3 = d_4$. The first case has only $D$ terms, each $O(1)$, in the sum over $d_k$. The second case is a bit more tricky. Assuming $d_1 = d_3$ w.l.o.g., by Cauchy-Schwarz and by the assumptions on correlations (note that the weights are i.i.d.,

so we can take the expectation w.r.t. the weights out of the sum),

$$\sum_{d_1,d_2} \mathbb{E}\left(X^i_{p_1} X^i_{p_2} X^i_{d_1} X^i_{d_2}\right)\left(X^j_{p_3} X^j_{p_4} X^j_{d_1} X^j_{d_2}\right) \mathbb{E}\left((W^*_{d_1} - W^0_{d_1})^2 (W^*_{d_2} - W^0_{d_2})^2\right) \tag{53}$$

$$\leq \sum_{d_1,d_2} \sqrt{\mathbb{E}\left(X^i_{p_1} X^i_{p_2}\right)^2} \sqrt{\mathbb{E}\left(X^i_{p_3} X^i_{p_4}\right)^2} \mathbb{E}\left(X^i_{d_1} X^i_{d_2}\right)^2 \mathbb{E}\left((W^*_{d_1} - W^0_{d_1})^2 (W^*_{d_2} - W^0_{d_2})^2\right) \tag{54}$$

$$\leq c\, D\, \sqrt{\mathbb{E}\left(X^i_{p_1} X^i_{p_2}\right)^2} \sqrt{\mathbb{E}\left(X^i_{p_3} X^i_{p_4}\right)^2} \mathbb{E}\left((W^*_1 - W^0_1)^2 (W^*_2 - W^0_2)^2\right). \tag{55}$$

Since other terms in the full sum are $O(1)$, the contribution of the second case is $O(D)$.

The last case is what we had for covariance. Since $d_1 = d_2$ and $d_3 = d_4$,

$$\sum_{d_1,d_3} \mathbb{E}\left(X^i_{p_1} X^i_{p_2} (X^i_{d_1})^2\right)\left(X^j_{p_3} X^j_{p_4} (X_{d_3})^2\right) \mathbb{E}\left((W^*_{d_1} - W^0_{d_1})^2 (W^*_{d_3} - W^0_{d_3})^2\right) \tag{56}$$

$$= D(D-1)\gamma^D_{p_1 p_2}\gamma^D_{p_3 p_4}\sigma^4_W + D\gamma^D_{p_1 p_2}\gamma^D_{p_3 p_4}\mathbb{E}\left(W^*_1 - W^0_1\right)^4. \tag{57}$$

Now we can combine all three terms, using that the weights have finite second and 4th moments. We will also use that for $d_1 = d_2$ and $d_3 = d_4$, we can ignore the second term with $\mathbb{E}\left(W^*_1 - W^0_1\right)^4$ since it's an $O(D)$ contribution to an $O(D^2)$ sum. Therefore,

$$\mathbb{E}\, R^2_{Ni} R^2_{Nj} = \frac{1}{D^2}\sum_{p_k}\left(\prod_{k=1}^4 \alpha_{p_k}\right)\left(\sigma^4_W D(D-1)\gamma^D_{p_1 p_2}\gamma^D_{p_3 p_4} + O(D)\right) \tag{58}$$

$$= \left(\sigma^2_W\, \alpha^\top \Gamma^D \alpha\right)^2 + O\left(\frac{1}{D}\right) = \sigma^4_R(D) + O\left(\frac{1}{D}\right). \tag{59}$$

**Asymptotic of the second moment.** To find the limit of the variances, we need to compute

$$\mathbb{E}\, \widetilde{R}^2_{Ni} = \frac{1}{D}\mathbb{E}\sum_{pp'd}\alpha_p \alpha_{p'}\, X^i_p X^i_{p'} (X^i_d)^2 (W^*_d - W^0_d)^2. \tag{60}$$

This variable is uniformly integrable since its 4th moment exists (almost identical computation to $R^2_{Ni}$) and is uniformly bounded for all $N$ (using Billingsley (1999), immediately after Theorem 3.5 for $\epsilon = 2$). Therefore, $\lim_N \mathbb{E}\, \widetilde{R}^2_{Ni} = \mathbb{E}\lim_N \widetilde{R}^2_{Ni}$ (Theorem 3.5 in Billingsley (1999)). The latter is easy to compute, since

$$\mathbb{E}\sum_d (X^i_d)^2 (W^*_d - W^0_d)^2 = D\, \sigma^2_W, \tag{61}$$

$$\mathbb{V}\text{ar}\left(\mathbb{E}\sum_d (X^i_d)^2 (W^*_d - W^0_d)^2\right) = \sum_{d,d'}\text{Cov}\left((X^i_d)^2 (W^*_d - W^0_d)^2, (X^i_{d'})^2 (W^*_{d'} - W^0_{d'})^2\right) \tag{62}$$

$$= O(D) + \sum_{d,d'\neq d}\text{Cov}\left((X^i_d)^2 (W^*_d - W^0_d)^2, (X^i_{d'})^2 (W^*_{d'} - W^0_{d'})^2\right) \tag{63}$$

$$= O(D) + \sigma^2_W \sum_{d,d'\neq d}\text{Cov}\left((X^i_d)^2, (X^i_{d'})^2\right) = O(D), \tag{64}$$

where in the last line we again used the assumption on summable covariances of $(X^i_d)^2$.

Therefore, by Chebyshev's inequality $\frac{1}{D}\sum_d (X^i_d)^2 (W^*_d - W^0_d)^2$ (note the returned $1/D$) converges to $\sigma^2_W$ in probability. Therefore, by Slutsky's theorem $\widetilde{R}^2_{Ni}$ converges to $\sigma^2_W \sum_{pp'}\alpha_p \alpha_{p'}\, X^i_p X^i_{p'}$. Since the expectation of $X^i_p X^i_{p'}$ is $\sigma_{pp'}$, we can denote $\Sigma_{\mathcal{A}}$ as a matrix with entries $\sigma^2_{pp'}$, such that

$$\mathbb{E}\, \widetilde{R}^2_{Ni} = \sigma^2_W\, \alpha^\top \Gamma^D \alpha \to \sigma^2_W\, \alpha^\top \Sigma_{\mathcal{A}} \alpha. \tag{65}$$

Since we assumed that $R_{Ni}$ converges to a random variable, and $R^2_{Ni}$ is uniformly integrable (since its 2nd moment exists), we can claim that $\mathbb{E}\, R^2_{Ni} \to \sigma^2_W\, \alpha^\top \Sigma_{\mathcal{A}} \alpha = \mathbb{E}\, R^2$.

For $R_{Ni}^2 R_{Nj}^2$, we can repeat the calculation. It's also uniformly integrable since its 8th moment exists: $\mathbb{E} \prod_{k=1}^{8}(W_{d_k}^* - W_{d_k}^0)$ leaves out $D^4$ terms out of $D^8$, the weights have a finite 8th moment, and the inputs are bounded. Therefore,

$$\mathbb{E} R_{Ni}^2 R_{Nj}^2 \rightarrow \sigma_W^4 \left(\alpha^\top \Sigma_{\mathcal{A}} \alpha\right)^2 . \tag{66}$$

**CLT for exchangeable arrays.** Assume that $\sigma_R^2(D) = O(1)$ (defined in Eq. (47)) has a non-zero limit, and $D$ is large enough for $\sigma_R^2(D) > 0$. Then $G_{Ni} = R_{Ni}/\sigma_R(D)$ is zero mean, unit variance and has a finite third moment. It also satisfies the following:

$$\mathbb{E} G_{N1} G_{N2} = O\left(\frac{1}{D}\right) = o\left(\frac{1}{N}\right) , \tag{67}$$

$$\lim_N \mathbb{E} G_{N1}^2 G_{N2}^2 = 1 , \tag{68}$$

$$\mathbb{E} |G_{N1}|^3 = O(1) = o(\sqrt{N}) . \tag{69}$$

Therefore, $G_{Ni}$ satisfies all conditions of Theorem 2 in Blum et al. (1958), and

$$\frac{1}{\sqrt{N}} \sum_{i=1}^{N} G_{Ni} \longrightarrow_d \mathcal{N}(0,1). \tag{70}$$

Therefore, we also have

$$\frac{1}{\sqrt{N}} \sum_{i=1}^{N} R_{Ni} \longrightarrow_d \mathcal{N}(0, \sigma_W^2 \, \alpha^\top \Sigma_{\mathcal{A}} \alpha). \tag{71}$$

(Note that in the case $\alpha^\top \Sigma_{\mathcal{A}} \alpha = 0$, we don't need to apply CLT since the sum converges to zero in probability by Chebyshev's inequality.) $\qquad\square$

Now we're ready to prove the main result (see the more informal statement in the main text, Theorem 1).

**Theorem 1.** In the assumptions of Lemmas 1 to 3, the scaled sequence of stochastic processes $\Xi^D = \left(\mathbf{X}\right)^\top \left(\mathbf{X}^D \mathbf{H}^D \left(\mathbf{X}^D\right)^\top\right)^{-1} \left(\mathbf{y}_D - \mathbf{y}_D^0\right)$ converges to a Gaussian process:

$$\widetilde{\Xi}^D = \frac{D \, \mathbb{E} \, \nabla^2 \phi^{-1}(W_d^0/\sqrt{D})}{\sqrt{N}} \Xi^D \rightarrow_d \mathcal{GP}(0, \sigma_W^2 \Sigma) , \tag{72}$$

where $\Sigma$ is the covariance matrix over $X_d$ and $\sigma_W^2 = \mathbb{E}\,(W_d^* - W_d^0)^2$.

**Remark.** Lemma 1 used the following correlation assumptions: $\sum_{d'=1}^{\infty} \left(\mathbb{E} X_d X_{d'}\right)^2 \le c$ and $\sum_{d'=1}^{\infty} \left|\text{Cov}\left(X_d^2, X_{d'}^2\right)\right| \le c$. Lemma 2 also used $\sum_{d'=1}^{\infty} \mathbb{E} X_d X_{d'} \le c$. In terms of Theorem 1 in the main text, the second condition corresponds to $c'_{ij}$. The first and the third conditions are joined into the $c_{ij}$ condition for convenience since it implies both conditions used here.

*Proof.* First, for an $\mathcal{A}$, $\alpha$-projection of the scaled stochastic process, we split the solution into two terms ($\mathbf{A}$ is defined in Eq. (24)):

$$\sum_{p \in \mathcal{A}} \alpha_p \, \widetilde{\Xi}_p^D = \frac{1}{\sqrt{N}} \sum_{p \in \mathcal{A}} \alpha_p \left(\mathbf{X}_{:p}\right)^\top \mathbf{A}^{-1}(\mathbf{y}_D - \mathbf{y}_D^0) \tag{73}$$

$$= \frac{1}{\sqrt{N}} \sum_{p \in \mathcal{A}} \alpha_p \left(\mathbf{X}_{:p}\right)^\top (\mathbf{y}_D - \mathbf{y}_D^0) + \frac{1}{\sqrt{N}} \sum_{p \in \mathcal{A}} \alpha_p \left(\mathbf{X}_{:p}\right)^\top \left(\mathbf{A}^{-1} - \mathbf{I}_N\right)(\mathbf{y}_D - \mathbf{y}_D^0). \tag{74}$$

We proved convergence of the first term to a Gaussian in Lemma 3. By Lemma 2, the second term converges to zero in probability. By Slutsky's theorem, since the first term converges to a Gaussian in distribution, and the second one to zero in probability, the whole expression converges to a Gaussian.

**Convergence to a Gaussian process.** We've shown that for any scalar weights $\alpha$ and a finite index set $\mathcal{A} \subset \mathbb{Z}_{>0}$, the projection of the sequence of the scaled stochastic processes $\widetilde{\Xi}^D$ w.r.t. $\mathcal{A}, \alpha$ converges to a Gaussian in distribution (with parameters found in Lemma 3). Therefore, $\widetilde{\Xi}^D$ converges to a Gaussian process in distribution by Lemma 6 of Matthews et al. (2018). $\qquad\square$

The teacher weights setup with $y_D = \sum_{d=1}^{D} \frac{1}{\sqrt{D}} X_d W_d^*$ simplifies the proofs, but it is not necessary. Here we provide a similar result for a generic $y$ without a detailed proof:

**Theorem 2.** In the assumptions of Lemmas 1 to 3, additionally assume that labels $y^i$ are zero-mean, bounded, and independent of $D$, $X_d^i$ and $W_d^0$, and also that $W_d^0$ are zero-mean. Then the scaled sequence of stochastic processes $\Xi^D = (\mathbf{X})^\top \left( \mathbf{X}^D \mathbf{H}^D (\mathbf{X}^D)^\top \right)^{-1} (\mathbf{y} - \mathbf{y}_D^0)$ converges to a Gaussian process:

$$\frac{D \, \mathbb{E} \, \nabla^2 \phi^{-1}(W_d^0/\sqrt{D})}{\sqrt{N}} \Xi^D \to_d \mathcal{GP}(0, (\sigma_y^2 + \sigma_{W^0}^2)\Sigma) \,, \tag{75}$$

where $\Sigma$ is the covariance matrix over $X_d$ and $\sigma_{W^0}^2 = \mathbb{E} \, (W_d^0)^2$, $\sigma_y^2 = \mathbb{E} \, y^2$.

*Proof.* The label setup doesn't affect Lemma 1. For Lemmas 2 and 3, due to the independence assumption for the labels and the weights, the proofs of the lemmas split into moment calculations for $W_d^0$ and $y$. The former is identical to the original calculation for $W_d^* - W_d^0$, since the weights are zero-mean. The latter doesn't involve sums over repeating coordinates, simplifying moment computations. The overlap is similar to the computation for $W_d^0$. With this change in the lemmas, the rest of the proofs is the same as for Theorem 1. $\square$

## A.1 Rich and lazy regimes

Assuming for simplicity that all initial weights are the same and positive ($w_d^0 = w^0 = \alpha/\sqrt{D}$) and that $\sigma_\lambda^2$ is equal to 1, then for $\xi \sim \mathcal{N}(0,1)$, Eq. (13) becomes

$$\nabla \phi(w^\infty) \approx \nabla \phi(w^0) + \frac{\sqrt{N}}{D \, \nabla^2 \phi^{-1}(w^0)} \, \xi \,. \tag{76}$$

For $p$-**norms**, $\phi(w) = \frac{1}{p} w^p$; $\nabla \phi(w^0) = (w^0)^{p-1}$; $\nabla^2 \phi^{-1}(w^0) = \frac{1}{p-1} (w^0)^{2-p}$. Therefore,

$$\nabla \phi(w^\infty) \approx \frac{\alpha^{p-1}}{D^{(p-1)/2}} + \frac{(p-1)\sqrt{N}\alpha^{p-2}}{D^{p/2}} \, \xi = \frac{\alpha^{p-2}}{D^{p/2}} \left( \alpha \, \sqrt{D} + (p-1) \, \sqrt{N} \, \xi \right) \,. \tag{77}$$

As long as $\alpha \gg \sqrt{N/D}$, the initial weights (first term) will be much larger than the weight change (second term), and we will be in the lazy regime. This additionally sets a limit on the dataset size. Put another way, for $\alpha = 1$ and the standard "lazy" initialization with $O(\sqrt{1/D})$ weights, the number of data points $N$ has to be much smaller than $D$ for the linearization to make sense (in addition to the more technical assumption on the $N$ scaling in Theorem 1).

For **negative entropy**, $\phi(w) = w \log w$; $\nabla \phi(w^0) = 1 + \log w^0$; $\nabla^2 \phi^{-1}(w^0) = w^0$. Therefore,

$$\nabla \phi(w^\infty) \approx 1 + \log \alpha - \frac{1}{2} \log D + \frac{\sqrt{N}}{\alpha\sqrt{D}} \, \xi \,. \tag{78}$$

Unlike for $p$-norms, the initial weights' contribution grows to infinity even as $\alpha$ tends to zero. As a result, we require only $\alpha \gg \sqrt{N/(D \log D)}$ to keep learning in the lazy regime. However, for $\alpha = 1/\sqrt{D}$ (i.e. $O(1/D)$ weights, the standard rich regime scaling) negative entropy still results in large weight changes since $N$ grows with $D$.

Overall, different potentials result in different asymptotic boundaries of the lazy regime. Negative entropy allows smaller initialization schemes while maintaining the lazy regime.

## B Experimental details

### B.1 Linear regression

For experiments in Section 4.1, we optimized models using mirror descent, where the gradients were computed in the standard SGD way with momentum of 0.9 and no weight decay (with mixed precision). Learning rate was changed during learning according to the cosine annealing schedule. The initial learning rate was chosen on a single seed via grid search over 16 points (log10-space

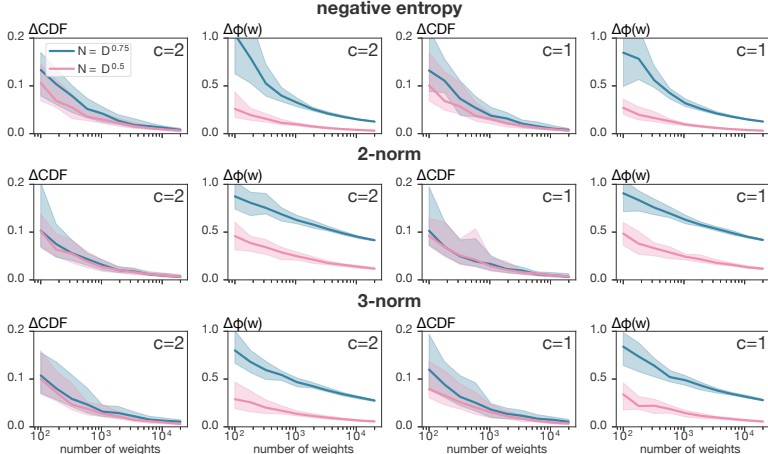

Figure 6: Linear regression solutions for negative entropy, 2-norm and 3-norm potentials. First and third columns: integral of absolute CDF difference ($\Delta$CDF) between normalized uncentered weights and $\mathcal{N}(0,1)$; second and third columns: magnitude of weight changes relative to the initial weights in the dual space ($\Delta\phi$). First two columns: quickly decaying correlation between $x_i$ ($c = 2$); last two columns: slowly decaying correlation ($c = 1$). All plots: 30 seeds, median value (solid line) and 5/95% percentiles (shaded area), $N = D^{0.5}$ (orange) and $N = D^{0.75}$ (blue).

from 1e-7 to 1e-1) and 100 epochs. If the minimum MSE loss was larger than 1e-3, we increased the number of epochs by 100 and repeated the grid search (up to 500 epochs; all grid searches converged to MSE smaller than 1e-3). The widths were chosen as 10 points in a log10-space from 1e2 to 2e4. The full results are shown in Fig. 6.

## B.2 POTENTIAL ROBUSTNESS

In Section 4.2, both figures in Fig. 3B were obtained in the following way. For $D = 100$ and 30 seeds, we sampled a 1000-dimensional weight sample $\mathbf{w}^0$ from either $\mathcal{N}(0, 1/D)$ or a log-normal with $\mu = -\log\sqrt{2D}$, $\sigma^2 = \log 2$ multiplied by a $\pm 1$ with equal probability (so it has the approximately the same variance as the Gaussian). We then computed the final weights $\mathbf{w}$ for the potential $\phi$ as $\mathbf{w} = \nabla\phi^{-1}\left(\nabla\phi(\mathbf{w}^0) + 0.1\,\nabla\phi(1/\sqrt{D})\,\mathcal{N}(0,1)\right)$, so the Gaussian change had a smaller magnitude than a weight with a standard deviation of $1/\sqrt{D}$.

## B.3 FINETUNING

In Section 4.3, for $N$ chosen datapoints, the dataset was randomly sampled from the 50k available points for each seed (out of 10). Networks were trained on the cross-entropy loss using stochastic mirror descent; the gradients had momentum of 0.9 but no weight decay, batch size of 256. Learning rate was changed during learning according to the cosine annealing schedule. The initial learning rate was chosen on a log10 grid of 5 points from 1e-5 to 1e-1. The initial number of epochs was 30, but increased by 30 up to 4 times if the accuracy was less than 100%. All resulting accuracies from this search were $\geq 99\%$. (Note that this is the desired behavior, since we're testing the models that fully learned the train set.) The dataset was not augmented; all images were center cropped to have a resolution of 224 pixels and normalized by the standard ImageNet mean/variance. We trained networks with mixed precision and FFCV Leclerc et al. (2022); for the latter we had to convert ImageNet to the FFCV format (see the accompanying code).

The magnitude of weight changes was usually smaller than 0.2, although for 2-norm for ResNet18/34 and CORnet-S, and 3-norm for EfficientNets b0,1,3 it reached 0.3-0.5. For negative entropy, the magnitude always stayed smaller than 0.02.

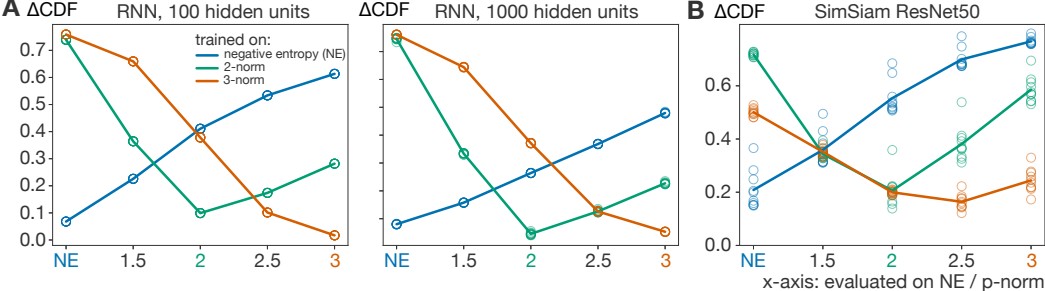

Figure 7: Same setup as Fig. 4B. **A.** Recurrent neural networks (RNNs) with 100/1000 hidden units trained and then finetuned on row-wise sequential MNIST. **B.** SimSiam ResNet50 finetuned with the same self-supervised loss.

We have conducted experiments with recurrent neural networks trained on row-wise sequential MNIST LeCun et al. (2010). The networks were trained on the train set, and then finetuned on a subset of the test set (same procedure as for deep networks) for $N = D^{0.5}$ (the number of weights scales quadratically with the hidden size, so $N$ equals the number of hidden units). The Gaussian fits, and non-Gaussianity of weight changes measured with a wrong potential were remarkably similar to our theoretical predictions and behavior of deep networks (Fig. 7A). Note that for networks trained with $p$-norms but evaluated with negative entropy, if the weight flips its sign during training we can immediately conclude that negative entropy was not used for training. Because we're finetuning already trained networks, most weights don't flip signs. We plot the distribution of changes in the dual space for all weights for simplicity (we can do it in the code since the dual weights are signed).

We also evaluated a self-supervised loss from SimSiam Chen & He (2021): for a pre-trained ResNet50 (see the full architecture in Chen & He (2021)), we finetuned it with the same self-supervised loss $l = 0.5\,\mathbf{z}_1^\top \mathbf{p}_2 + 0.5\,\mathbf{z}_2^\top \mathbf{p}_1$ for positive pairs 1/2 passed through decoders with or without stop-gradient ($\mathbf{z}$ and $\mathbf{p}$). We trained the network for 50 epochs and used the learning rate that minimized the loss the best, since we don't evaluate accuracy. We evaluated weight changes in the encoder (a ResNet50). The results were very similar to networks finetuned with a supervised loss, confirming our results for supervised experiments (Fig. 7B).

## B.4 COMPARISON TO SYNAPTIC DATA

In Section 4.4, we used the data from Dorkenwald et al. (2022). We used the the accompanying dataset, specifically the "spine_vol_um3" data available here. The dataset contained 1961 spine volume recordings.

For negative entropy fitting (Fig. 5A), we sampled the initial weights as $10^{-1.42}$ with probability $p = 0.77$ and $10^{-0.77}$ with $p = 0.23$. In the dual space, we added a centered Gaussian with $\sigma = 10^{-0.23}$. The means and probabilities exactly match the ones fitted in Dorkenwald et al. (2022) (Table 2); the standard deviations were fitted as $10^{-0.24}$ and $10^{-0.22}$; we used a log-average since the standard deviations were very similar and we expect the Gaussian parameters to be the same across weights.

For 3-norm (Fig. 5B), we obtained a qualitative fit. With equal probability, the initial weights were either 1e-2 or log-normal with $\mu = -2.7$ and $\sigma = 0.9$. In the dual space, the Gaussian change was zero-mean with $\sigma = 0.0012$. A small part of the final weights became negative with this sampling procedure, which we treated as synapses that were reduced to zero weight during learning and therefore we excluded them from the plots.

The 3-norm approximately fits the data because for a small constant initial weight $c$, the final weight will be approximately log-normal: $\log\left(\nabla\phi^{-1}(\nabla\phi(c) + \xi)\right) \approx \log\left(c + \xi/\nabla^2\phi(c)\right) \approx \xi/\nabla^2\phi(c)$. The second part of the mixture is already log-normal and is barely changed by the Gaussian change, resulting in a mixture of two (approximately) log-normals.

## C  NOTES ON MIRROR DESCENT

For negative entropy, we assumed that the weights do not change signs. This way, $|w| \log |w|$ is strictly convex for either $w > 0$ or $w < 0$ and the gradient of potential is invertible. This is implied by the exponentiated gradient update Eq. (7), which indeed preserves the signs of the weights.

Powerpropagation Schwarz et al. (2021) reparametrized network's weights $\mathbf{w}$ as $\mathbf{w} = \theta \, |\theta|^{\alpha-1}$, such that gradient descent is done over $\theta$. The original weights are therefore updated for a loss $L$ as (assuming all terms remain positive for simplicity):

$$\mathbf{w}^{t+1} = \left( \theta^t - \eta \frac{\partial L}{\partial \mathbf{w}^t} \frac{\partial \mathbf{w}^t}{\partial \theta^t} \right)^{\alpha} \approx (\theta^t)^{\alpha} - \alpha \, (\theta^t)^{\alpha-1} \, \eta \, \frac{\partial L}{\partial \mathbf{w}^t} \frac{\partial \mathbf{w}^t}{\partial \theta^t} \tag{79}$$

$$= (\theta^t)^{\alpha} - \alpha^2 \, (\theta^t)^{2\alpha-2} \, \eta \, \frac{\partial L}{\partial \mathbf{w}^t} = \mathbf{w}^t - \alpha^2 \, \eta \, (\mathbf{w}^t)^{\frac{2(\alpha-1)}{\alpha}} \frac{\partial L}{\partial \mathbf{w}^t} \, , \tag{80}$$

where we linearized the update in the first line and then used the relation between $\theta$ and $\mathbf{w}$. If we treat $(\mathbf{w}^t)^{\frac{2(\alpha-1)}{\alpha}}$ as the Hessian of $\phi^{-1}$ in mirror descent (see Eq. (14)), powerpropagation can be viewed as an approximation to mirror descent with $p$-norms, where $p = 2/\alpha$. For $\alpha = 2$, we can also treat powerpropagation as an approximation to exponentiated gradient (Eq. (7)).

