# OpenReview forum: "Synaptic Weight Distributions Depend on the Geometry of Plasticity"
_ICLR.cc/2024/Conference — ICLR 2024 spotlight_

### Official Review · Reviewer_mbP3 · 2023-10-25

**Soundness:** 1 poor
**Presentation:** 3 good
**Contribution:** 2 fair
**Rating:** 1
**Confidence:** 4

**Summary:**

In this work, the authors use mirror descent principles to derive the distribution of final synaptic weights, under certain assumptions. Theoretical findings demonstrate that this distribution is determined by mirror descent potentials. Analyzing synaptic weight distributions before and after training is crucial for understanding brain mechanisms. The paper applies mirror descent theory to distinguish learning rules in the brain, modeling it under chosen synaptic geometry. Drawing from this theory, the paper shows that, under specific assumptions, weight changes in dual space follow a Gaussian distribution. This insight aids in inferring synaptic geometry from the weight distribution. The authors validate this approach through experiments on artificial neural networks and demonstrate its applicability to real neural data.

**Strengths:**

The application of concepts from mirror descent to underscore the significance of selecting the appropriate distance function is a commendable idea.

**Weaknesses:**

The main drawback is the multitude of assumptions made in the article, leading to a highly robust conclusion. I've outlined what I consider to be the most unreasonable five assumptions:

1. Our brain is a feedforward network with no feedback and lacks any dynamic processes.
2. During learning, only the neurons in the last layer of our brain update.
3. Our brain should exclusively perform a supervised learning task.
4. Our brain uses gradient descent algorithms.
5. The optimal values for our brain's neurons for a given problem are unique and deterministic.

Each of these five assumptions is overly restrictive, and some are evidently incorrect, conflicting with known experimental evidence.

Moreover, the conclusion itself also appears highly unreasonable: the distribution of synaptic strength in our brain neurons, as well as the brain's structure and the environmental context (task execution), appear to be unrelated. The distribution of synaptic strength is solely related to geometry.

 And the conclusion itself lacks significance. What matters is, assuming the article's conclusion is correct, why we would choose this specific geometry, what advantages it offers, and how we arrive at such a choice.

**Questions:**

see Weaknesses

---

> ### Author Response · Authors · 2023-11-16
> **Response 1/2**
>
> We are pretty disappointed with this review, as it is clear that the reviewer did not actually read our paper carefully, nor try to give constructive or insightful critiques.
>
> First, the reviewer raises concerns about **five** assumptions we *supposedly* make in this paper, but **only one of them** is made, and then only partially (see item 4, below). It leads us to wonder if it is just a simple misunderstanding, (in which case that may be a result of poor communication on our part), or whether Reviewer mbP3 was simply not engaged enough with the paper to notice that we do not make the assumptions they claim we do. We agree that each of these five assumptions listed here is overly restrictive, and some are evidently incorrect, conflicting with known experimental evidence. But, as noted, we **do not** make these assumptions. Below, we explain this for each of the claimed assumptions we make:
>
> > “Our brain is a feedforward network with no feedback and lacks any dynamic processes.
>
> We don’t assume that the network is feedforward, in fact, in the submission **we present results with recurrent neural networks, namely CORnet (a recurrent architecture)**. We have also added RNN experiments to the revised paper.
> > During learning, only the neurons in the last layer of our brain update.
>
> **We never state this or make any assumptions this way**. This comment is therefore very wide of the mark and indicates the reviewer has not engaged with the paper. Please see section 2.1 Deep networks onwards.
> > Our brain should exclusively perform a supervised learning task.
>
> **We don’t assume this**. We only assume that the brain is engaged in a task that can be framed as regression, but we would point out that **self-supervised learning and reinforcement learning also often rely on various forms of regression**, thus there is nothing about the assumption of regression that forces us into supervised learning. However, we agree that verifying our theoretical statement that our analysis is loss agnostic is valuable, and **we have included experiments with a self-supervised loss here to demonstrate this point**.
> > Our brain uses gradient descent algorithms.
>
> We do assume that an ANN trained by gradient-based methods can be used as a working model for the brain, and as a working abstraction, can be used to understand aspects of how the brain learns and works. This does not mean we are committed to the assumption that the brain explicitly calculates gradients and uses them for its updates. Rather, it means we assume that the updates are non-orthogonal to the gradient. Notably, this is a mathematical necessity if weight updates are small, because otherwise performance would not improve, see e.g. Raman, D. V., Rotondo, A. P., & O’Leary, T. (2019). Fundamental bounds on learning performance in neural circuits. Proceedings of the National Academy of Sciences, 116(21), 10537-10546. Notably,  this framework is widely used in computational neuroscience literature because it allows researchers to make tractable theoretical advances that can generate biological hypotheses - as we do in this manuscript. The implications of this assumption are stated clearly and we believe this work contributes to an established literature which precisely aims at elucidating the presence of gradient-following plasticity in the brain.
> > The optimal values for our brain's neurons for a given problem are unique and deterministic.
>
> This is possibly the strangest of the reviewer’s concerns, as we never state this anywhere and actually explicitly assume the opposite – that **the network should be overparameterized**, which would lead to there being many different possible solutions that would be arrived at depending on various stochastic factors, such as initialization and data sampling. Thus, again, the reviewer appears to have not really engaged with the paper, and possibly, not even read it beyond the abstract.

---

> ### Author Response · Authors · 2023-11-16
> **Response 2/2**
>
> > Moreover, the conclusion itself also appears highly unreasonable
>
> Here we would like to flag this as a highly unscientific comment. We are not sure what to say other than to encourage the reviewer to channel their incredulity into an objective assessment of our theoretical and experimental evidence, as well as our work’s novelty. We encourage them to read the other reviewer’s comments to challenge their initial opinion. Reviewing scientific papers should not consist of comments along the lines of “No way, I can’t believe it”, but rather, careful assessment of the evidence provided. We are surprised we need to clarify this here, but it seems we do.
>
> Regarding the claim that our conclusion is highly unreasonable, **we’d like to point out that the data collected on synaptic weight distributions in the brain seems to support our conclusions. Notably, the distribution of synaptic spine sizes across brain regions and species [Buzsáki and Mizuseki, 2014] is highly robustly log-normal**. That suggests that regardless of the architecture or function of a neural circuit, the synaptic weight distribution is log-normal, which could easily be explained if the synaptic geometry is the major determining factor, and this was preserved across species and brain regions. Thus, far from being unreasonable, our conclusion is both supported by our theory and experiments, and existing data in the neuroscience literature. We will be sure to make this point in any revisions.
>
> > And the conclusion itself lacks significance.
>
> We strongly disagree, and must highlight again that the reviewer is making oddly subjective comments with no argument to back up their claims. Why is this conclusion insignificant? If we are correct, and changes in the weight distribution will be dominated by the synaptic geometry, then that would give us an experimental means of assessing synaptic geometry, which would be a major insight into how the brain works! We also think the reviewer's surprise regarding our conclusions are indicative of actual significance. If science were never surprising, it would be unnecessary. We do agree that future work is required for this theoretical contribution to fully leave its mark on AI and computational neuroscience, but we think this an exciting and novel first step.
>
> In conclusion, we were very disappointed with this review. Not because it is a low score, but **because it lacks any actual engagement with our paper, both in terms of the theory we develop and the evidence we present**. It is very frustrating as an author to have a reviewer make various false claims about assumptions we make, then dismiss our conclusions based on the reviewer’s apparent inability to believe the conclusions without any discussion of what the theory or data presented shows.
>
> We hope the AC will ignore this review when rendering their decision.

---

### Official Review · Reviewer_GuwS · 2023-10-27

**Soundness:** 4 excellent
**Presentation:** 4 excellent
**Contribution:** 3 good
**Rating:** 8
**Confidence:** 4

**Summary:**

This paper provides a theoretical framework for understanding the geometry of synaptic changes based on the synaptic weight distribution. In particular, they show that the synaptic weight distribution in real brains are inconsistent with vanilla gradient descent, which assumes Euclidean distances in weight space. Based on the Mirror descent framework, they derive a theorem that shows that the distribution of weights after learning depends on the initial distribution plus a Gaussian in the dual space for large models and data. The authors then show that the theorem holds in a linear regression setting, and that hypothesizing the wrong potential yields non Gaussian weight distribution. Then, the authors show that their theorem holds well despite unsatisfied assumptions in the case of classic deep learning convnets fine-tuned on ImageNet data. Finally, they consider biological data and conclude that both the distribution before and after learning are needed to conclude about the geometry of synaptic changes, but that vanilla SGD can be ruled out.

**Strengths:**

The strengths of the paper are:

**Originality:** The approach is original and principled, and it can make verifiable predictions about the biology. It is nice that the theory is robust to changes in the potential.

**Quality:** The quality of the text and figures is high.

**Clarity:** The paper stays very clear despite the theorem being a tad math-heavy.

**Significance:** I think tackling the question of what geometry is followed by synapses in the brain is significant. It is true that most computational neuroscience work assume vanilla SGD so the question whether the resulting weight distributions are coherent with biological data is an important point.

**Weaknesses:**

The main weakness of the paper is that even though the theory is agnostic to the loss function and dataset, it seems that it is not agnostic to the architecture given section 4.3. So it would maybe make sense to also try more plausible architectures than ANNs to test the theory like continuous Hopfield networks or Spiking networks trained with surrogate gradients for instance, or at least vanilla RNNs, since the brain is highly recurrent. This would provide a sense of how important is the architecture vs the learning rule.

**Questions:**

- Section 4.4: I find the choices of the initial $w_0$ surprising to demonstrate the fits, isn't $w_0$ supposed to be the distribution before learning? If yes then it is surprising that it needs to be a mix of two constants since it is likely not the case in brains. Could the authors elaborate on that?

- Can you elaborate on the link between geometry of plasticity and the locality of the learning rule? I would expect non Euclidean geometry to be non local since the metric tensor needs to be inverted, however Eq 7 does look local.

- Isn't the use of optimizers such as Adam a way to have a different geometry of plasticity in ANNs?


Minor:

Fig 3B: caption is not coherent: top/bottom should be left/right.

Fig 4: I see the histograms in pink and not blue as written in the caption.

---

> ### Author Response · Authors · 2023-11-16
>
> Thank you for the review!
>
> We agree that architecture choices play a role in our experiments. However, for all architectures, we’ve observed a strong contrast between multiplicative (NE) and additive (2-norm) plasticity rules. The last two architectures, CORnet-S and CORnet-RT, had several recurrent blocks; we included them to test if recurrence can break our conclusions. In the updated pdf (see Appendix B.3), **we included new RNN experiments** on row-wise sequential MNIST; the new results are very consistent with our prediction. (We agree it would be valuable to try e.g. spiking networks, but we would leave it to future work.)
>
> > Section 4.4: I find the choices of the initial w_0 surprising to demonstrate the fits
>
> Apologies for an unclear message! We wanted to provide the simplest initialization that would lead to the final weight distribution to illustrate how just knowing the final distribution is not enough. That plot could also be reproduced if we started from two Gaussian in the log space (same means as for the point initialization). A simple example would be that data itself: if we treat the recorded weight distribution as the initial conditions, and train a bit more with the negative entropy potential, the resulting weight distribution should have two slightly wider Gaussian in the log space.
>
> > Can you elaborate on the link between geometry of plasticity and the locality of the learning rule?
>
> If $\nabla\phi(\mathbf{w})_i=f(w_i)$ (so, a function of a single weight and not others), the update becomes local since it doesn’t depend on other weights. This is true for the potentials we discuss. However, for Euclidean geometry, we additionally need $f(w_i) = w_i$. For negative entropy, we get $f(w_i) =1 + \log w_i$, so the primal-dual weight mapping stays local but is no longer Euclidean. We mention this (albeit more briefly) just before Eq. 14, but have now expanded that explanation.
>
> > Isn't the use of optimizers such as Adam a way to have a different geometry of plasticity in ANNs?
>
> It would depend on the optimizer. A simple momentum (with scalar, but possible adaptive, parameters), wouldn’t change mirror descent solutions (Gunasekar et al. (2018)). Adam or similar optimizers might change the space of solutions, but they usually try to make gradient updates unit-variance, which intuitively should strengthen Gaussian convergence. See the added discussion after Th. 1 in Sec. 3.
>
> > minor points
>
> Thank you, we have corrected them.

---

> > ### Comment · Reviewer_GuwS · 2023-11-21
> > **Good rebuttal**
> >
> > Thanks for answering my points and providing additional experiments. I remain confident in my assessment and still recommend accept.

---

### Official Review · Reviewer_DFTN · 2023-10-31

**Soundness:** 4 excellent
**Presentation:** 4 excellent
**Contribution:** 2 fair
**Rating:** 5
**Confidence:** 2

**Summary:**

The paper uses the mirror descent framework to derive results for the distribution of weights in converged networks. The paper shows that under a linearity assumption (and an L2 loss), the final distribution of weights is Gaussian, and this result can be used to analyse weight updates in trained deep nets and also synaptic updates in brain regions.

**Strengths:**

Strengths:
1. The paper is, for the most part clearly written, and understandable. The mathematical sections in the main paper are also quite accessible and easy to follow.

2. The idea for using the mirror descent to analyse synaptic weight changes is an elegant one, and the simplifying the analysis using the linearity assumption indeed seems to make the problem more tractable for use on networks applied to data.

3. The experiments in section 4 are fairly thorough in validating the theory presented in section 3.

**Weaknesses:**

1. The distinction between "loss function" and "potential": The paper claims in section 2 that the mirror descent framework splits a synaptic weight update gradient into two terms: one dependent on "external errors" i.e. the loss function $l$, and one "intrinsic to the synapse" i.e. the potential $\phi$. However, I am skeptical that this is generally true: to the best of my understanding, we can say that gradients with respect to the potential $\phi$, $\Delta\phi$ are independent of gradients wrt the the loss $\Delta l$. However, given that both the loss function and potential are functions of the synaptic weights, and that the loss function is either chosen by the practioner or unknown (in the case of the brain), unless we _explicitly_ choose $l$ to represent _only_ errors extrinsic to the synapse, it is not clear to me how we can guarantee the above statement. Relatedly, it is also not clear to me whether we can always pick a loss function that only captures non-synapse-related changes in weights, and if we can guarantee that we have managed this in every use case.

2. Finally, the paper claims in section 1 to "make experimental predictions" and provide "new theoretical insights...about learning algorithms in the brain". In section 4.4, while it is clear from the results and the analysis that a non-Gaussian distribution of weights under the linearity assumption might indicate a non-Euclidean synaptic geometry, it is not clear how we can interpret this for insights about how learning happens in the brain, and for potential plasticity mechanisms. It is also not clear how extensible the analogy is from deep recurrent networks is to the brain -- synaptic weight changes may not be analogous to network weight updates via gradient descent, and therefore any theoretical results based on gradient descent may not port easily to neuroscientific insights.

The weaknesses taken together seem to make the paper fall short of the claims in the introduction. Without the interpretability, and further experimentation with neuroscience data, it is not clear to me what the framework adds in terms of neuroscientific insight. While it might be useful for analysing deep net behaviour, again, it is not clear how this can be used to improve deep net training either.

Minor point:
Terminology -- the terms "loss function", "distance" and "geometry" are used interchangeably without clarification, throughout the paper, and particularly in sections 1-2.1. It is not until section 2.1 that the distinction between the three terms and what they indicate becomes somewhat clear. This hinders readability and generally makes it very hard to grasp the premise of the paper or its implications without these terms being clearly explained.

**Questions:**

1. How can we guarantee that the loss function / potential function separation truly represents a separation between extrinsic and intrinsic factors in the weight changes?

2. How to evaluate whether the linearity / Gaussian assumption has been violated, in cases where the loss and potential function are unknown?

3. How to make the "geometry of synaptic weight changes" more interpretable for neuroscientific insight?

---

> ### Author Response · Authors · 2023-11-16
> **Response 1/2 (questions)**
>
> Thank you for your review, the points you have raised have been useful for us. We address weakness 1 and question 1 together below. Regarding weakness 2, please see our response to question 3, and we also understand it to overlap with the applicability of gradient-based learning to neuroscience (shared with reviewer jXnh). Here’s a slightly shortened response:
>
> First, our work uses gradient descent to construct a rigorous theory. However, **our result ultimately relies on the large-sum behavior of update terms**. If we replace the gradient $\nabla l$ wrt the loss in Eq. 5 $\nabla\phi(w^{t+1})=\nabla\phi(w^{t})-\eta\nabla l$, with anything else that drives learning, like a 3-factor Hebbian rule, the intuition would still hold: a large sum of small and mostly independent updates looks Gaussian in the dual space.
>
> More generally, we assume that an ANN trained by gradient-based methods can be used as a working model for the brain, and as a working abstraction can be used to understand aspects of how the brain learns and works. As long as the learning algorithm the brain uses descends a gradient, even if it is never explicitly calculated, then this framework can be applied. However, this does not mean that we are committed to the brain doing gradient descent, per se. But it is a commonly applied framework that allows researchers to make tractable theoretical advances that can generate biological hypotheses - as we do in this manuscript.
> > How can we guarantee that the loss function / potential function separation truly represents a separation between extrinsic and intrinsic factors in the weight changes?
>
> The motivation of the paragraph just before Sec. 2.1 was to convey a way to think about credit assignment in terms of the mirror descent framework we present in Sec. 2. **We are not committed to this separation of factors per se in biology. However, we think it can be helpful to think in terms of this separation.** Extrinsic credit assignment factors are those external to the synapse, such as a neuron-wide error signal multiplied by presynaptic activity (which corresponds to the gradient). These external factors are then used to update the synapse (somehow), which is the intrinsic step. These factors are not necessarily separated in real neurons, however, within the mirror descent framework they are, and the intrinsic step is captured by the potential. We hope this clarifies this point and are happy to discuss it further. We have updated our communication of this point in the manuscript.
> > How to evaluate whether the linearity / Gaussian assumption has been violated, in cases where the loss and potential function are unknown?
>
> Sec. 3.1 indirectly addresses this point, in short, whether the loss and/or potential function are unknown does not impact the assessment of linearity /Gaussian assumption violation:
> 1. Linearity: the norm of the weight changes should be ~ an order of magnitude smaller than the weights in the dual space. This comes from applying a 1st order Taylor series approximation, and can be verified for a candidate potential.
> 2. Gaussian assumption: this could be broken by correlations in inputs. However, we think this is unlikely, because this is a common feature in natural sensory data [Ruderman (1994)] and neural activity [Schulz et al. (2015)] (we mention this just before Sec. 3.1). We also empirically investigate the robustness of this violation in the deep network experiments (Figs ).
> We will strengthen our communication of these points in the main text.
>
> > How to make the "geometry of synaptic weight changes" more interpretable for neuroscientific insight?
>
> By “geometry of synaptic weight changes” we mean the set of properties that describe the relationship between different synaptic weight configurations. In this paper we focus on distance, i.e. which synaptic configurations are “far” or “close” to each other for a neuron. We highlight that training models with gradient descent implicitly assumes a Euclidean distance assessment of which synaptic weight configurations are close to or far from each other. However, this does not fit with neurobiology, for example synapses cannot change sign! As such, Euclidean distance does not make sense from a neuroscientific perspective because a weight change that involves a sign change is not penalized more than one that does not. We therefore focus on alternative distance functions (Bregman divergences, which are parameterized by a potential). These distance functions induce different weight change geometries, as they change which weight configurations are deemed far or close to one another.
>
> In this sense, the true synaptic geometry of the brain is currently unknown. But our work enables a direct experimental proposal – if we (1) pick a task, (2) measure the weights before and after task learning, we can then (3) find the potential (and therefore geometry) that best fits a Gaussian change in the dual space (see Fig 1 for primal/dual space distinction).

---

> ### Author Response · Authors · 2023-11-16
> **Response 2/2 (scope)**
>
> While our approach needs some further development for direct use with neural data, as per points that you raised, **we’d also like to emphasize the scope of our contributions to neuroscience as we see them:**
> 1. This work is a significant step towards aligning biological synaptic weight changes with network weight updates via gradient based methods. **The results we present are both novel and technically non-trivial, and lay the foundation for such further analysis and theory**. For example, determining the theoretical limits of the Gaussian approximation to the true weight change (see appendix) coupled with empirical tests with deep networks.
> 2. We show that, despite the common use of basic gradient descent as a plasticity model in computational neuroscience, it’s not consistent with log-normality, multiplicative updates, and Dale’s law. **Our framework allows us to address this without re-evaluating various neuro-inspired models of how the brain might implement backpropagation** (abstaining from whether this is the right approach or not).
> 3. The ANN & gradient descent approach, despite its recent popularity, is often only vaguely linked to biology. One of the problems is a lack of experimental applicability. **Here we make a step in this direction: going from theory and deep learning-driven models to experiment design and predictions**. Namely that we can experimentally determine the synaptic geometry of the brain.
>
> > While it might be useful for analysing deep net behaviour, again, it is not clear how this can be used to improve deep net training either.
>
> There is recent work showing that the use of different p-norms as potentials can result in better performance than gradient descent (which is the 2-norm) [Sun et al., 2023]. Furthermore, a weight re-parameterisation that is equivalent to the negative entropy potential in certain settings, was found to facilitate training sparse networks and be beneficial for continual learning [Schwarz et al., 2019] (see also Appendix C). Finally, in RL, mirror descent with geometries has long been used for bandit tasks [Shalev-Shwartz, 2011]. Therefore an exciting future direction is to explore the differences and benefits between gradient descent and a weight update algorithm derived from estimating biological synaptic geometry.  If the reviewer updates their score and the paper is accepted for publication we will add a discussion of this subject to the appendix or discussion, as we agree it is an important topic to communicate.
>
> We hope we have clarified your questions and addressed your concerns about our neuroscientific contributions. If not, we would very much welcome further discussion.

---

> > ### Comment · Reviewer_DFTN · 2023-11-18
> > **Thank you, and some follow-up questions**
> >
> > I thank the authors for the detailed responses and for the updates to the manuscript -- particularly in clarifying assumptions and intuition in Sections 2 and 3.
> >
> > Since the authors responded to the questions in my review in detail, I have increased my score to a 5.
> >
> > I do have some follow-up questions:
> >
> > > We are not committed to this separation of [intrinsic and extrinsic] factors per se in biology. However, we think it can be helpful to think in terms of this separation.
> >
> > Thank you for the clarification, and for adding this to the manuscript. Could you elaborate on how you think this separation could be useful, if we are unable to extend this assumption to neurons in every case?
> >
> > To clarify: to the best of my understanding, the paper asserts that the mirror descent framework is useful because (a) we have this separation between loss and potential updates, and (b) it allows us (under linearity assumptions in the dual space) to assume a Gaussian distribution for the final synaptic weights. From the experiments in 4.4, we can at least say something about synaptic weights in real neuronal networks, given (b). It is still not clear to me how the same is true of (a).
> >
> >
> > > Gaussian assumption: this could be broken by correlations in inputs [...] (we mention this just before Sec. 3.1). We also empirically investigate the robustness of this violation in the deep network experiments (Figs )
> > > [...] determining the theoretical limits of the Gaussian approximation to the true weight change (see appendix) coupled with empirical tests with deep networks.
> >
> > Thank you for the clarification, and the additions to section 3. However, once again, I am struggling to understanding the neuroscientific implications of these results. For example, the last lines of Section 3 (before 3.1) state "However, for generic error terms we cannot rely on Eq. (8) and can only provide generic conditions for Guassian behavior. Therefore, while our work leverages gradient-based optimization, the intuition we present applies more broadly." -- it is not clear what this means. Which intuitions apply more broadly? -- that as long as weight updates are _approximately_ independent, we can still assume a final Gaussian distribution? What happens when this assumption breaks too?
> >
> > In addition, the appendix (I assume section A) is quite dense, and difficult to parse -- it is not clear how this section defines the limits of the Gaussian approximation, and again, how this affects neuroscientific interpretation. Would the authors be able to summarize this here or in the main paper?
> >
> > > [If] the paper is accepted for publication we will add a discussion of this subject to the appendix or discussion, as we agree it is an important topic to communicate.
> >
> > Thank you: including this in the discussion would be great.
> > Also in general, from reading the responses here and going over the additions to the manuscript, my main takeaway is this: The manuscript claims to be bridging theoretical and experimental predictions, but sections 2-4 (and in particular explanations of the intuition and results) are theory-heavy, and do not allow the reader to easily infer the connection of these results to experimental/neuroscientific insights.
> >
> > For example, at the end of section 4.3, the paper states "Taken together, these empirical results suggest that Theorem 1 holds more generally and can be used to infer the distribution of weight changes beyond linear regression". While this makes it clear how the results relate to the derivations in section 2, it is not immediately clear what implications it has for experiment design or predictions.  Similarly, at the end of section 3.1 "This implies that for the same weight initialization and dataset size, negative entropy would typically produce smaller updates. Additionally, for the standard for deep networks initialization with [...]" presents intuition purely in terms of theory without drawing a clear connection to experiments / learning in the brain.
> >
> > I recognize that the discussion period is too short to make more substantial changes to the manuscript, but if accepted, I would recommend a re-write of these sections to make the connection more explicit and clear.

---

> > > ### Author Response · Authors · 2023-11-20
> > > **Response 1/2 (main text)**
> > >
> > > Again, we sincerely thank the reviewer for their helpful feedback. This discussion has been very useful for improving the paper. We also thank the reviewer for their thoughtful consideration in adjusting their score, despite the follow up questions. We hope we can clarify these now:
> > >
> > > > Could you elaborate on how you think this separation could be useful, if we are unable to extend this assumption to neurons in every case?
> > >  > ... (a) we have this separation between loss and potential updates, and (b) it allows us (under linearity assumptions in the dual space) to assume a Gaussian distribution for the final synaptic weights. From the experiments in 4.4, we can at least say something about synaptic weights in real neuronal networks, given (b). It is still not clear to me how the same is true of (a).
> > >
> > > Thank you for continuing to push us on this! Yes, you are correct that (at least currently) we cannot say anything that links the separation of the loss gradient and potential in mirror descent to a biological separation in real neurons.
> > >
> > > Addressing (a), if the underlying biological mechanism violates the mirror descent potential/error signal separation, we can’t apply this framework. This is definitely possible – however, most of the existing theories for error-driven learning (gradient descent, approximations to it, 3-factor Hebbian rules, etc.) do factorize!
> > >
> > > From a neuroscience perspective, in this work we provide evidence against the brain updating synaptic connection strengths via gradient descent because we prove that gradient descent is not consistent with the log-normal weights seen in the brain. While interesting and important (we think), there are also many other reasons why the brain cannot be implementing gradient descent - at least similar to how ANNs do it. Indeed how the brain might approximate gradient descent is active area of research, e.g.  Lillicrap et al. (2016); Liao et al. (2016); Akrout et al. (2019); Podlaski & Machens (2020); Clark et al. (2021). As such, there is ongoing work and many papers exploring how the brain might be calculating error terms and approximating gradients.
> > >
> > > Because of the factorisation, our work does not run counter to this pre-existing body of research. Instead, we are able to leverage these studies for the “extrinsic” error term which is independent of the “intrinsic” potential (resulting in distance) term that we focus on. From our perspective this is very important, because the factorisation can be used to make all of the previous literature on credit assignment (even) more biologically plausible.
> > >
> > > We hope this is clear? In summary we are enthusiastic about the factorisation from a theoretical perspective because it explicitly allows us to not re-evaluate all of the previous work studying how the brain might estimate the extrinsic error term!
> > > Again thank you for pushing us to be clearer about this point. We have again updated the paragraph in section 2, and will revisit this for a camera ready version.
> > >
> > > > Therefore, while our work leverages gradient-based optimization, the intuition we present applies more broadly." -- it is not clear what this means. Which intuitions apply more broadly? -- that as long as weight updates are approximately independent, we can still assume a final Gaussian distribution? What happens when this assumption breaks too?
> > >
> > > Yes, without gradient descent and the application of Theorem 1, as long as weight updates are approximately independent, we can still assume a final Gaussian distribution. This comes from a large sum behavior: $\nabla\phi(w^\infty)=\nabla\phi(w^0) + \sum_t g^t$, so we can expect $\sum_t g^t$ to follow the central limit theorem (CLT; by approximate independence assumption). If we measure the change with a different potential, that sum will be non-linearly transformed, and so CLT would not apply.
> > > When this independence assumption breaks, then unfortunately we won’t be able to say much. But, this assumption is likely to hold in neural systems since noise is very prevalent in biology .
> > >
> > > Extending the discussion past an answer to your question:
> > > Perhaps it would help to clarify how we use Theorem 1. If we denote $\nabla\phi=f_1$ and another potential $f_2$, we’re saying that $f_1(w^\infty)=f_1(w^0) + \mathcal{N}$. Then if we measure the change with $f_2$, we get $f_2(w^\infty)=f_2(f_1^{-1}(f_1(w^0) + \mathcal{N}))$. If $f_1$ and $f_2$ are similar, then we can linearize $f_2(w^\infty)$ and also see something Gaussian. If not, the change will be non-Gaussian. Therefore we are able to use Gaussianity in the dual space to determine the correct potential. **We have updated the manuscript after the theorem to communicate this point (Section 3).**

---

> > > > ### Author Response · Authors · 2023-11-20
> > > > **Response 2/2 (appendix A summary)**
> > > >
> > > > > Would the authors be able to summarize this [Appendix A] here or in the main paper?
> > > >
> > > > Yes! The very brief summary is provided in the proof sketch after Th. 1. As any Gaussian convergence, we’ll end up using the central limit theorem (although an unusual version of it) wrt the dataset size. With this in mind, the steps are:
> > > > 1. First, we decide how we take the large data & network limits both to infinity. We therefore have a vector of weight changes whose size grows to infinity. To discuss what it converges to, we consider the vector of weight changes to be part of a stochastic process (so, an already infinite-dimensional vector). We then consider the convergence of this stochastic process to some limiting process (in our case, to a Gaussian process). This then allows us to approximate a sample of weight changes as a sample from that limiting process.
> > > > 2. [Lemma 1] Next, we show that the inverse matrix in the weight changes expression behaves like an identity matrix. If we just took the limit wrt network size, this would be trivial since off-diagonal elements are products of independent vectors. But we also increase the matrix size, so we need a more careful deviation bound. This is where we get (the most of) the requirements for how the dataset size should scale with network size.
> > > > 3. [Lemma 2] shows that the deviations resulting from Lemma 1 don’t affect the final weight change expression when we take the limits.
> > > > 4. [Lemma 3] shows that if you replace that inverse matrix with an identity, you can apply the exchangeable central limit theorem to the weight change expression. At this point, it’s done for any finite slice of that stochastic process we defined. The exchangeable central limit theorem has much more restrictive moment conditions than the standard CLT, since it doesn’t require independence, so we have to carefully check all of them.
> > > > 5. [Theorem 1] Finally, we combine the lemmas and show that any finite slice of that stochastic process converges to a Gaussian. Due to the properties of Gaussian processes, we can conclude that the whole process converges as well.
> > > >
> > > > We thank the reviewer for their acknowledgment of the restricted time-period. If accepted we will provide a summary similar to the above in the appendix (perhaps a shorted one due to space constraints). We also acknowledge your point that sections 2-4 are theory heavy and connecting these sections to neuroscience insights is not easy. For a camera-ready version will certainly take the time to go over these sections with this in mind. Our apologies for not doing so now, in light of how long there is left for discussion we decided to prioritize responding to your above questions. We hope we have been successful in clarifying them!

---

### Official Review · Reviewer_jXnh · 2023-11-06

**Soundness:** 3 good
**Presentation:** 3 good
**Contribution:** 3 good
**Rating:** 8
**Confidence:** 4

**Summary:**

This paper develops a theory, based on mirror descent, of the distribution of synaptic weights changes. In particular, this distribution depends on the geometry of synaptic plasticity. They test theory predictions which are largely verified.

**Strengths:**

I’m a big fan of the ambition in this paper. It’s original, the results are good, and this line of work could have big implications. That being said, I found the paper really lacking in clarity (see below).

**Weaknesses:**

The big assumption that the brain is doing gradient descent…

“Notably, even if the brain does not estimate gradients directly, as long as synaptic weight updates are relatively small, then the brain’s learning algorithm must be non-orthogonal to some gradient in expectation” What’s the actual citation for this?? (i.e. not a review)

I found the paper pretty dense, and not easy to follow what was going on where, and I always had to keep lots of things in mind at any moment. I’m not sure exactly what to suggest, but considerable rewriting / putting things in appendices / focussing on intuition would be a good idea.

Couldn’t parse Fig 3. Needed more help in the caption / annotation..

“Nevertheless, with this data we can rule out a Euclidean synaptic geometry, and if we do have access to w0, then our results show that it is indeed possible to experimentally estimate the potential function.” Where do you show this? It’s not in Fig 5.

I didn’t follow the explanation before eqn 14. Could do with some clarifying.

**Questions:**

See weaknesses

---

> ### Author Response · Authors · 2023-11-16
> **Response 1/2 (gradient descent)**
>
> Thank you for the review!
> > The big assumption that the brain is doing gradient descent…
>
> Our result only partially relies on gradient descent. We need it, along with the result in Eq. 8, to have a rigorous theory for Gaussian behavior during learning. However, as we outline in the second paragraph after Eq. 7, **our result ultimately relies on the large-sum behavior of update terms**, and these updates need not come from gradient descent. So, if we replace the gradient $\nabla l$ wrt the loss in Eq. 5 (the mirror descent update), $\nabla\phi(w^{t+1})=\nabla\phi(w^{t})-\eta\nabla l$, with anything else that governs learning, like a 3-factor Hebbian rule, the intuition would still hold: a large sum of small and mostly independent updates looks Gaussian in the dual space, $\nabla\phi(w^{\infty})=\nabla\phi(w^{0})  + \sum_{t=1}^\infty g^t\approx \nabla\phi(w^{0}) +\mathcal{N}$. But such a general case would make theory much harder (unless we put some trivial assumptions like literal independence of updates). Gradient-based optimization leads to the Bregman divergence result, which in turn leads to a closed-form approximate solution, which we can show to be Gaussian. We have added this explanation to Sec. 3.
>
> More generally, we assume that an ANN trained by gradient-based methods can be used as a working model for the brain, and as a working abstraction can be used to understand aspects of how the brain learns and works. As long as the learning algorithm the brain uses descends a gradient, even if it is never explicitly calculated, then this framework can be applied.  However, this does not mean that we are committed to the brain doing gradient descent, per se. But it is a commonly applied framework that allows researchers to make tractable theoretical advances that can generate biological hypotheses - as we do in this manuscript.
>
> We also offer another point of view: if backpropagation is taken as the working model of the brain, is it consistent with neural data? Our conclusions are that it’s not in the simplest form (gradient descent) due to the solutions it finds, but other variants such as exponentiated gradient are consistent (with log-normals, Dale’s law, and multiplicative updates). This is important and impactful given the number of computational papers that assume gradient descent as a working model of learning in the brain.
>
> Lastly, we note that there is a sizable literature exploring learning in the brain under the assumption of gradient-guided plasticity. The question about the validity of this assumption is important – and work like ours help make progress– but this larger question is beyond the scope of our paper.
>
> > What’s the actual citation for this?? (i.e. not a review)
>
> We referred to this: Richards and Kording. The study of plasticity has always been about gradients. The Journal of Physiology, 2023.
>
> But the basic mathematical idea is that if you’re slowly improving your performance with small weight changes, you must be approximately following the gradient (wrt performance) since it is the mathematical direction for performance improvement. If you were not, then there would be no improvement in loss. As an additional citation (non-review) for this point, see Raman, D. V., Rotondo, A. P., & O’Leary, T. (2019). Fundamental bounds on learning performance in neural circuits. Proceedings of the National Academy of Sciences, 116(21), 10537-10546.

---

> ### Author Response · Authors · 2023-11-16
> **Response 2/2 (writing)**
>
> > I found the paper pretty dense, and not easy to follow what was going on where, and I always had to keep lots of things in mind at any moment.
>
> Thank you for this comment! We agree that we had to condense a lot of things in the main text, partly because we draw on a range of prerequisites: mirror descent is not a well known topic in theoretical neuroscience.
>
> We’d be happy to expand the intuition/re-write in some parts, and would appreciate guidance re which sections. We’re assuming that the densest part is Sec. 3 (the main result): would expanding on Lagrange multipliers from the beginning help? Additionally, would making the theorem statement more informal help?
>
> **Currently, we expanded the discussion before Th. 1 and compressed Sec. 3.1 (see the updated pdf)**. But we’re also hesitant to put all technical details to the appendix since we are already hiding significant theoretical contributions there to prioritize readability!
>
> > Couldn’t parse Fig 3
>
> Left plots: with more weights (lazy regime), the weight changes become more Gaussian (CDF difference) and smaller in magnitude. Right plots: if the change is Gaussian for one potential, measuring it with the wrong potential would result in a very non-Gaussian weight change. We hope this is clear?
>
> > Where do you show this? It’s not in Fig 5.
>
> Apologies, we should’ve made it more clear. We can rule out Euclidean geometry since it can’t produce Gaussian changes in the log space. We can estimate the geometry by finding the potential with the most Gaussian change (if the weights before and after learning are known).
>
> > I didn’t follow the explanation before eqn 14. Could do with some clarifying.
>
> We agree there are few steps that we had to condense due to space constraints here. There is a linearization step and a local potentials discussion. We have expanded the derivation for the former in the text. Re local potentials: If $\nabla\phi(\mathbf{w})_i=f(w_i)$ (so, a function of a single weight and not others), the update becomes local since it doesn’t depend on other weights. This is true for the potentials we discuss, but not e.g. for $\phi(\mathbf{w})=\|\mathbf{w}\|_2$ (non-squared norm) since the gradient would depend on other weights.

---

> ### Author Response · Authors · 2023-11-20
> **Updated writing**
>
> As the discussion period is coming to an end, we have further updated the manuscript after our discussion with reviewer DFTN. Their comments also focused on the clarity of the paper and we think our responses and changes are applicable to your comments.
>
> Thank you again for your positive and helpful review, we hope the changes we have made to the manuscript have improved its clarity and as such we hope you would consider a score increase.

---

> > ### Comment · Reviewer_jXnh · 2023-11-23
> > **Thanks for responses**
> >
> > Many thanks for your efforts in making the paper more presentable. I have increased my score as, while I admit that I have some reservations, I think this sort of paper is really useful for the community.

---

### Author Response · Authors · 2023-11-16
**Summary of changes/new experiments**

We would like to thank the reviewers for their helpful comments. In light of the reviewer’s suggestions, we have modified the text and the content of the paper (the changes are shown in red).

**Summary of changes**
1. We improved the clarity of Sec. 2 and Sec. 3 (before Theorem 1).
2. We expanded the intuitive description of our result to explain how it can work for any learning mechanism, but doing theory would become harder (see the changes after Theorem 1 in Sec. 3).
3. We compressed Sec. 3.1 (lazy/rich scaling), moving the derivation to the Appendix A.1.
4. (Edit 2) We added more clarifications to Sec. 2 and Sec. 3. Sec. 3.1 moved to a paragraph. Sec. 4.2 shortened since it's now partially covered in Sec. 3.

**Summary of experiments**:
1. We ran additional experiments with RNNs and further confirmed our results are robust to architecture choice.
2. We also ran an experiment with a self-supervised loss (ResNet50 + SimSiam) to show that our theory applies to unsupervised settings too.
See Appendix for both: Sec. B.3 and Fig. 7. We believe these new results significantly strengthen the paper.

Please see the individual comments for reviewer specific responses.

---

### Meta-Review · Area_Chair_yyMK · 2023-12-11

**Metareview:**

This paper derives a theoretical link between the (synaptic) weight distributions in neural network and the geometry of the synaptic weight changes (distance). Using the framework of mirror descent, the work shows that these weight distributions are Gaussians for the potential function that is consistent with the correct geometry. Feasibility was demonstrated in the context of DNN but also in the analysis of synaptic weight distributions measured in BNNs (biological neural networks). The results show that synaptic plasticity in the brain is not evaluated in an Euclidean distance as traditionally assumed by gradient descent model. The proposed method has the promise to allow us to experimentally determine the true geometry of synaptic plasticity in the brain.

The general idea to investigate the 'geometry of synaptic plasticity' and the technical sophistication of the proposed approach and method were generally considered substantial strengths of the work. A certain lack of clarity in description was criticized, which is not surprising given that the paper is quite technical and dense. Furthermore, certain concerns regarding the practical application to study the synaptic geometry of biological neural networks were raised but mostly convincingly addressed during the discussion period.

One reviewer stood out with a very low score (1) and a minimal review that provided only a generally formulated, negative assessment. The AC mostly agrees with the authors' rebuttal where they show that the negative review is not based on factual accuracy. Unfortunately, the reviewer did not elaborate on his criticism nor replied to the authors rebuttal. As such the AC acknowledges the reviewer's rather diffuse concerns but will strongly reduce their impact on the overall assessment of the paper.

**Justification For Why Not Higher Score:**

The strength of the paper (besides its topic being interesting) is its level of technical sophistication. However, this also implies that there is a small but non-zero probability that there may be somewhat hidden assumptions that could limit the practical application of the method.

**Justification For Why Not Lower Score:**

It is a technically sophisticated paper that addresses an interesting and important topic. Gradient descent with its implicit Euclidean geometry is still a fairly popular learning algorithm for networks intended to mimic biological brains (see work e.g. by Yamins et al.). Being able to determine the actual 'geometry of plasticity' based on synaptic weight distributions may potentially unlock additional insights into the power of biological neural networks.

---

### Decision · Program_Chairs · 2024-01-16

Accept (spotlight)